# Probabilistic Flood Inundation Mapping through Copula Bayesian Multi-Modelling of Precipitation Products

Francisco Javier Gomez[1]; Keighobad Jafarzadegan[1], Hamed Moftakhari[1], and Hamid Moradkhani[1]

[1]Department of Civil, Construction and Environmental Engineering, Center of Complex Hydrosystems Research, The University of Alabama, 35487 Tuscaloosa, US

*Correspondence to*: Francisco J. Gomez (fjgomez1@crimson.ua.edu)

**Abstract.** Accurate prediction and assessment of extreme flood events are crucial for effective disaster preparedness, response, and mitigation strategies. One crucial factor influencing the intensity and magnitude of extreme flood events is precipitation. Precipitation patterns, particularly during intense weather phenomena such as hurricanes, can play a significant role in triggering widespread flooding over densely populated areas. Traditional flood prediction models typically rely on single source precipitation data, which may not adequately capture the inherent variability and uncertainty associated with extreme events due to certain limitations in precipitation generation framework, availability or both spatial and temporal resolutions. Moreover, in coastal regions, the complex interaction between local precipitation, river flows and coastal processes (i.e., storm tide) can result in compound flooding and amplify the overall impact and complexity of flooding pattern. This study presents an implementation of Global Copula-embedded Bayesian Model Averaging (BMA) (Global Cop-BMA) framework for improving the accuracy and reliability of extreme flood modelling. The proposed framework integrates a collection of precipitation products with different spatiotemporal resolutions to account for uncertainty in forcing data for hydrodynamic modelling and generating probabilistic flood inundation maps. The methodology is evaluated over Hurricane Harvey, a catastrophic weather event characterized by intense precipitation and compound flooding processes over the city of Houston in the state of Texas in 2017. The results show a significant improvement in predictive accuracy compared to those based on a single precipitation product (e.g., NSE performance of single QPE are in the range of 0.695 to 0.846 while the Cop-BMA yields a NSE of 0.858), demonstrating the merits of the Global Cop-BMA approach. Furthermore, the research extends its impact by generating probabilistic flood extension maps that account not only for the primary influence of precipitation as a flood driver but also for the intricate nature of compound flooding processes in coastal environments.

## 1. Introduction

The inherent uncertainty associated with hydrodynamical modelling, exacerbated by the complex and often non-linear relationships, presents a challenge to accurately predict extreme flood events (Jafarzadegan et al., 2023). This uncertainty is frequently linked to diverse categories of errors encompassing inputs, such as the resolution and availability of topobathymetric data (Alipour et al., 2022; Liu and Merwade, 2018; Savage et al., 2016), as well as the quality and precision of boundary

conditions derived from hydrological models, other type of hydraulic/hydrodynamic models, or extracted from hydrometric measurements at monitoring stations (Abbaszadeh et al., 2019, 2022b; Jafarzadegan et al., 2021a, b; Oruc Baci et al., 2023). Beyond these factors, additional sources of uncertainty arise from inherent errors within numerical models, including the type and dimensions of the model, governing equations, assumptions, simplifications of physical processes, and the construction of the computational domain (Bates, 2022; Liu et al., 2019; Teng et al., 2017).

Bayesian Model Averaging (BMA) has been used in the past two decades as a statistical framework for improving the reliability of hydrological or meteorological models by quantifying and reducing uncertainties arising from different models (e.g., Duan et al., 2007; Han & Coulibaly, 2017; Parrish et al., 2012; Raftery et al., 2005). BMA enables the incorporation of multiple model predictions, each possessing its own strengths and limitations, into a unified probabilistic framework. Through this process, BMA techniques provide a robust means of generating ensemble predictions that not only capture the inherent variability of the system but also account for model uncertainties, parameter uncertainties, and data uncertainties. BMA applications have expanded into other domains, such as flood inundation models, aiming to achieve more accurate estimation of flood extent and water level while accounting for different sources of uncertainty during flood events (Huang & Merwade, 2023; Liu & Merwade, 2018, 2019; Moftakhari et al., 2017).The main limitation of BMA in hydrological applications, , is the use of same marginal distributions in the construction of joint probabilities, and that it is generally assumed that the data and the conditional PDF of the data follow a Gaussian distribution. Copula-embedded Bayesian Model Averaging (Cop-BMA) represents an advancement, distinguishing itself from the traditional BMA formulation, by constructing the joint distribution independently of the marginal distributions of the individual variables of analysis (Madadgar et al., 2014). This distinction positions Cop-BMA as a more reliable tool for considering uncertainty from the marginal distribution of the analysed data.

With the advancements in computational modelling, novel tools have emerged to optimize and enhance outcomes while incorporating new variables into the analysis. The incorporation of precipitation data directly into hydrodynamic models via Rain-on-grid (RoG) functionality stands among the innovative features that is gaining recognition by hydrodynamic modelers by allowing the incorporation of spatiotemporally varied precipitation data into the computational domain. Among various hydrodynamic models, the Hydrologic Engineering Center's River Analysis System (HEC-RAS) developed by the United States Army Corps of Engineers (USACE, 2022). It has the capability to simulate flooding conditions allows in both 1D and 2D. Although some investigations have explored the integration of RoG into the HEC-RAS 2D hydrodynamic model and assessed its performance (Costabile et al., 2020; David and Schmalz, 2021; Zeiger and Hubbart, 2021), a significant gap remains in comprehensively exploring the utility of RoG in result evaluation, comparisons with analogous computational models, and the analysis of uncertainties generated from its incorporation as a boundary condition. Currently, multiple regional and global precipitation data and products are available, exhibiting a wide range of spatial and temporal resolutions. These valuable data assets offer the opportunity to enhance the accuracy of hydrodynamic flood modelling to higher levels of detail, although, incorporating this type of information introduces an additional layer of uncertainty, prompting the need to account for these variations to enhance the accuracy of estimating both the extent and depth of flooding.

Comparisons of various precipitation products have been integral in the assessment of Quantitative Precipitation Estimation (QPE) techniques, particularly within the context of precipitation generation and its subsequent impacts. These evaluations encompass an array of data sources, such as observations from satellites, ground-based gauges, radar measurements, reanalysis products, and combinations thereof, all contributing to the nuanced understanding of precipitation patterns (e.g., Gavahi et al., 2023; Nelson et al., 2016; Wootten & Boyles, 2014). In addition to these comparisons, studies have researched the details of QPE techniques and products during extreme hydrometeorological events. The case of Hurricane Harvey serves as a prime example (Brauer et al., 2020; Gao et al., 2021; Habibi et al., 2021; Omranian et al., 2018). This event exhibited the importance of accurate precipitation estimation, given its critical role in extreme flooding. However, the differences between observed and derived precipitation values emphasize the presence of inherent errors and biases within precipitation products. Consequently, relying solely on one dataset for QPE could potentially lead to an incomplete representation of the complex conditions encountered during such extreme events (Gavahi et al., 2023).

The impact of Hurricane Harvey was deeply felt along the Texas coastline. It brought with it an approximate accumulated precipitation of over 1500 mm in the vicinity of Beaumont, TX, and resulted in estimated losses of 125 billion dollars based on the 2017 Consumer Price Index (Blake and Zelinsky, 2018) Given the significance of this hurricane and the widespread damage it caused across the state of Texas, considerable efforts have been undertaken to model and quantify the extent and depths of the flooding it generated. Various approaches, including numerical hydrodynamic models (Huang et al., 2021; Jafarzadegan et al., 2021a; Muñoz et al., 2022; Noh et al., 2019; Saksena et al., 2020; Sebastian et al., 2021; Stephens et al., 2022; Valle-Levinson et al., 2020; Wing et al., 2019), as well as combinations of different methodologies or type of models have been employed (Chen et al., 2021, 2022; Dullo et al., 2021).

By combining hydrodynamic modelling results driven with different precipitation datasets, Bayesian multi-modelling techniques have the potential to account for uncertainties in precipitation products and enhance the flood inundation mapping skills. This article presents an approach that incorporates both deterministic and probabilistic methods in the study of Hurricane Harvey event. On the deterministic front, the numerical results of the HEC-RAS 2D 6.3.1 hydrodynamic model, incorporating RoG, are evaluated to best describe the hydrodynamic behaviour of rivers, coastal and floodplain processes with a computationally affordable model. In parallel, a probabilistic approach is employed to use eight distinct precipitation products as forcing data to the hydrodynamic model to estimate an ensemble of flood extent and water depth in response to this hurricane-induced flood event. The deterministic approach provides a single representation of flood extents and depths based on predefined inputs and parameters, offering a clear understanding of potential inundation scenario evaluated. However, it fails to adequately capture the uncertainty associated with flood modelling, potentially leading to underestimation or overestimation of flood extents in other scenarios considering highly sensitive input parameters, which can impact the accuracy of results (Di Baldassarre et al., 2010; Bates et al., 2004).

Probabilistic flood inundation mapping incorporates probabilistic techniques to assess and quantify uncertainty, providing a more comprehensive understanding of the range of potential flood outcomes and associated risks. It allows the integration of

different datasets and input values, accommodating a wider range of initial and boundary conditions, and improving the robustness of flood predictions (Merwade et al., 2008; Di Baldassarre et al., 2010). Often this approach requires conducting numerous simulations to assess parameter uncertainty, leading to a substantial consumption of computational resources.

Consequently, there is a preference for utilizing models that make substantial flow assumptions to conduct these simulations more efficiently and reduce computational cost.

Overall, this study aims to 1) investigate the impacts of different precipitation data in the simulation of extreme floods, such as hurricane Harvey using HEC-RAS 2D and 2) quantify the uncertainties associated with different precipitation products by generating probabilistic flood inundation maps using the Global Copula Bayesian (Global Cop-BMA) multi-modelling

technique.

## 2. Methods

The methodology employed in this study centers on numerical hydraulic modelling and the assessment of flood extent and water elevation using the Global Copula Bayesian (Global Cop-BMA) multi-modelling technique. Fig. 1 represents the main steps required for the implementation of the proposed methodology. First, the HEC-RAS 2D hydrodynamic model is set up,

incorporating data such as roughness, boundary conditions (discharges, water levels, and precipitation), and terrain. In this step, the HEC-RAS 2D model is driven with different precipitation products to generate a collection of flood inundation maps. Second, the Cop-BMA technique is employed to combine the flood maps and produce a single probabilistic flood inundation map that accounts for the uncertainties associated with different precipitation products.

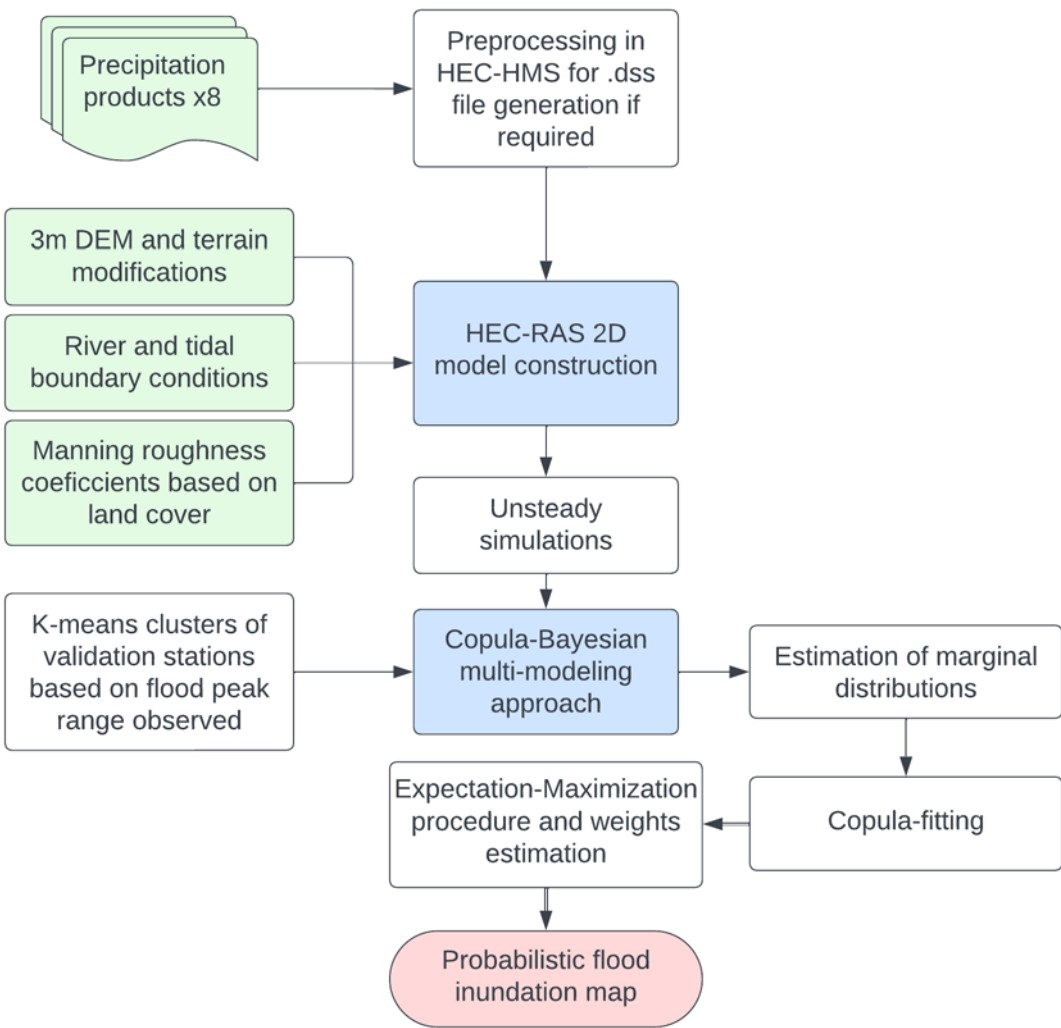

**Figure 1. Flowchart generated of the proposed methodology for probabilistic flood inundation mapping.**

### 2.1 Hydrodynamic Modelling

Flood extent and depth maps are typically obtained by performing 1D or 2D hydrodynamic modelling that numerically solves the Saint-Venant or Shallow Water Equations respectively. Each of these models possesses its own advantages and limitations in terms of computational complexity, assumptions of flow nature, practicality, accuracy, and precision (Bates, 2022; Teng et al., 2017). Among these options, 2D models offer a notable compromise, enabling flood modelling with a satisfactory level of detail while maintaining a manageable computational cost compared to their 3D counterparts. Furthermore, as compared to

1D models, they facilitate the calculation of water levels across floodplains in a more intricate and physically plausible manner over complex geometries.

While a variety of 2D models, both open-source and commercially licensed exist, the current study utilizes the HEC-RAS 2D model version 6.3.1. This choice is motivated by HR2D's open accessibility and significant improvements, such as the integration of subgrid concepts for mesh refinement and the incorporation of Shallow Water Equations (SWE). These enhancements mark a distinct advancement over previous versions, making HR2D a suitable candidate for flood modelling. Notably, it surpasses its predecessors, which were employed in studies involving Hurricane Harvey's impact on the city of
Houston (Garcia et al., 2020; Jiang et al., 2023; Scotti et al., 2020).

The hydrodynamic model setup is based on three primary inputs: the terrain, the roughness associated with land cover and land use types, and the boundary conditions or external forcings (typically discharge and/or water levels). Recent advancements in model capabilities have enabled the integration of additional boundary conditions within the computational domain. This integration enhances the physical representation of the system which results in more accuracy and reduces the reliance on other
types of models, such as hydrological models. In numerous flooding scenarios, precipitation plays a key role as a substantial portion of this flood driver transforms into direct runoff leading to flood inundation. This phenomenon is typically referred to as the pluvial impact of flooding and is particularly evident in events like Hurricane Harvey (Saksena et al., 2020). Hence, the RoG functionality within HR2D emerges as a pivotal feature to be incorporated into the methodology.

**2.2 Copula Bayesian Multi-Modelling Approach**

Among different multi-modelling approaches, Bayesian Model Averaging (BMA) has been widely used for combining multiple model predictions and producing more reliable results that account for the uncertainty of each model. BMA produces a predictive probability distribution function (PDF) of a variable, water surface elevation in this case, which is the weighted average of the PDFs associated with each model prediction. The weights reflect the prediction skill of different models. By considering the performance of all independent $k$ model predictions $[M_1, M_2, ... M_k]$, BMA eliminates the need to select a
single "best" model, thereby providing a more robust prediction (Madadgar and Moradkhani, 2014). The law of total probability is used to calculate the distribution of target (predicted) variable $y$ using both observed data and model predictions. Considering the dynamic nature of these models, the time component is integrated in the law of total probability as expressed in equation 1:

$$p(y^t|M_1^t, M_2^t, ..., M_k^t, Y) = \sum_{i=1}^{k} p(M_i^t|Y) \cdot p(y^t|M_i^t, Y) = \sum_{i=1}^{k} w_i \cdot p(y^t|M_i^t, Y) \tag{1}$$

$$\sum_{i=1}^{k} w_i = 1 \tag{2}$$

where $p(y^t|M_i^t, Y)$ is the PDF of $y^t$ given the model $M_i^t$ and training data $Y$ and $p(M_i^t|Y) = w_i$ is the likelihood of model prediction being corrected, given the observations, $Y$, during the analyzed period. These weights reflect the performance of models in predicting the target variable with a total sum equal to one. In summary, the weight $w$ reflects the degree to which a model aligns with the observed data; that is models demonstrating high-performance receive higher weights.

With BMA, the assumption that the posterior distribution is following a Gaussian distribution is commonly used as $p(y^t|M_i^t, Y) \sim g(y^t|M_i^t, \sigma_i^2)$, but this may not be correct in all cases given the nature of data used. In these cases, it is convenient to transform the data from its original space to a Gaussian space via Box-Cox transformation. Considering the target variable as water surface elevation, the Yeo-Johnson power transformation is preferred to account for negative values. This is particularly relevant in coastal environments where such values are commonly observed due to tidal conditions.

To overcome the limitations of BMA associated with the Gaussian distribution of variables and their joint distribution, a second solution involves integrating Copula multivariate functions into the BMA approach, known as Cop-BMA. Copulas are functions in the unit cube, which can link multi-dimensional distributions to their one-dimensional marginals (Sklar, 1959), they provide a flexible and powerful tool for modelling the dependency structure between variables, regardless of their individual marginal distributions and model dependency. This is particularly valuable in scenarios where the relationships

between variables are complex and may not follow a simple linear pattern. Cop-BMA modifies the BMA predictive distribution through relaxing the assumption on parametric posterior distribution $g(y^t|M_i^t, \sigma_i^2)$ replaced with a group of multivariate copula functions. Multiple copula functions have been applied to postprocess hydrological forecasts (Abbaszadeh et al., 2022a; He et al., 2018; Madadgar et al., 2014; Madadgar and Moradkhani, 2014), and are used in this study for the estimation of water surface elevation posterior distribution. Equation 1 is modified to incorporate copula functions replacing the posterior

distribution $p(y^t|M_i^t, Y)$ following the procedure from Abbaszadeh et al. (2022a). Supported by Sklar theorem copulas can express the joint behaviour among correlated variables through their marginal CDFs in equation 3.

$$P(x_1, \ldots, x_n) = C[P(x_1), \ldots, P(x_n)] = C(u_1, \ldots u_n) \tag{3}$$

where $C$ is the Cumulative Distribution Function (CDF) of the copula and $P(x_i)$ is the marginal distribution of $x_i$ denoted as $u_i$ for the interval [0, 1]. Using the PDF of copula, the joint probability density function of the variables involved can be

defined as follows:

$$p(x_1, \ldots, x_n) = c(u_1, \ldots, u_n) \prod_{i=i}^{n} p(x_i) \tag{4}$$

The conditional probability distribution of $x_1$ given $x_2$ is defined in equation 5:

$$p(x_1|x_2) = \frac{p(x_1, x_2)}{p(x_2)} \tag{5}$$

Considering the copula joint probability from equation 4, the equation 5 can be expressed as:

$$p(x_1|x_2) = \frac{p(x_1, x_2)}{p(x_2)} = \frac{c(u_1, u_2) \cdot p(x_1) \cdot p(x_2)}{p(x_2)} = c(u_1, u_2) \cdot p(x_1) \tag{6}$$

Since $u_1$ and $u_2$ are the observations $(y^t)$ and simulations $(M_k^t)$ respectively, the posterior distribution in equation 1 is replaced with the conditional probability distribution from equation 6 as:

$$p(y^t|M_1^t, M_2^t, \ldots, M_k^t, Y) = \sum_{i=1}^{k} w_i \cdot p(y^t|M_i^k, Y) = \sum_{i=1}^{k} w_i \cdot c\left(u_{y^t}, u_{M_i^t}\right) \cdot p(y^t) \tag{7}$$

Where $c(u_{y^t}, u_{M^t})$ represents the PDF of the copula function. To estimate weight $w_i$, it is required to maximize the log

likelihood function of the vector of parameter $\theta = \{w_i, i = 1, \ldots, k\}$ as:

$$l(\theta) = \log\left(\sum_{i=1}^{k} w_i \sum_{t=1}^{T} c\left(u_{y^t}, u_{M_i^t}\right) \cdot p(y^t)\right) \tag{8}$$

The Expectation-Maximization (EM) algorithm, proposed by Raftery et al. (2005), is used to maximize equation 8. This is achieved through iterative updates of the weights by adjusting a latent variable until a specified tolerance criterion is met.

In order to probabilistically estimate the flood extent and depth over a large domain, a comprehensive approach is necessary to spatially characterize the outcomes derived from different numerical models or, in this context, various hydrodynamic simulations with different precipitation products. This becomes especially crucial when the variables used in this study, namely the precipitation products and water level resulting from numerical simulations, exhibit significant spatial variability. Parameter regionalization plays an important role in identifying clusters or regions where assigning a single parameter for the whole domain is not reasonable (Jafarzadegan et al., 2020). To estimate weights for these clusters or regions, a global extension of the Cop-BMA has been developed, following the same procedure as the EM algorithm introduced earlier for the estimation of weights and likelihood (Yan et al., 2020). Likelihood function (equation 8) is adjusted to consider multiple stations over each cluster.

$$l(\theta) = \sum_{n=1}^{N} \log \left( \sum_{i=1}^{k} w_i \sum_{t=1}^{T} c \left( u_{y^t}, u_{M_i^t} \right) \cdot p(y^t) \right) \tag{9}$$

where $N$ refers to the number of stations per cluster.

## 3. Study Area and Data

The Galveston Bay area is located in southeastern Texas on the Gulf Coastal Plain and covers parts of Brazoria, Chambers, Galveston, Harris, and Liberty counties. As the largest estuary in the state, it exhibits a notable level of urbanization in the western zone, primarily attributed to the city of Houston. The city has several bayous and creeks that flow mostly southeastward into Galveston Bay. To the north is the San Jacinto River, which flows from the discharge of Lake Houston spillway to the south.

### 3.1 Model setup

The HEC-RAS 2D model is built through the RAS Mapper tool version 6.3.1 with Shallow Water Equations with Eulerian-Lagrangian Method (SWE-ELM) formulation for governing equations It has a total geometry extension of 5514.8 km$^2$ with 396,063 computational cells with spatial resolution of 200 x 200 meters refined to 75 x 75 meters or less in Houston city area (Garcia et al., 2020; Scotti et al., 2020). The unstructured meshing approach used in this study results in proper characterization of terrain complexities in urban areas while maintaining a reasonable computational time. For unsteady flow analysis in HR2D setup, an hourly simulation time window is defined between August 16/2017 to September 3/2017. The 2D flow domain is defined considering the most significant discharge contributions to the Galveston Bay area (Figure 2). The main highways in the Houston area, including Texas 8 Beltway and Interstate 610, serve as critical watershed boundaries for hydrodynamical

modelling in the urban regions. Therefore, an additional major effort was made to incorporate break lines along these features in Houston. This allows for proper hydro-enforcement and enhances hydraulic connectivity between the computational cells.

The NCEI Continuously Updated Digital Elevation Model (CUDEM) Bathymetric and Topographic DEM, with a 1/9 arc-second resolution(National Centers for Environmental Information, 2014) is used as the topography data. Since a fraction of the study area is highly urbanized, and there is no information on all the bridges, culverts, and geometry of the artificial channels. Topographic adjustments are made within RasMapper to guarantee and preserve the hydraulic characteristics of the streams.

Manning roughness coefficients are spatially assigned using the 2019 National Land Cover Database (Dewitz and U.S. Geological Survey, 2021). To reduce the spatial complexity of various land covers in the study area, the land cover map is simplified into five groups of developed/urban areas, forests/wetlands, open water, navigational areas, and barren land (crops, pasture, agriculture). In a previous research conducted by Muñoz et al. (2022), they used Latin Hypercube Sampling and tested various Manning roughness values for different land cover categories during Hurricane Harvey event. We use their calibrated parameters as a reference for HR2D model setup. These values are slightly adjusted during the calibration period, 7 days before the occurrence of Hurricane Harvey. It is worth mentioning that our simulations were performed on a desktop computer with an Intel Core i7-7700 CPU @ 3.60GHz and 32GB RAM memory, averaging about seven hours per simulation for the time window.

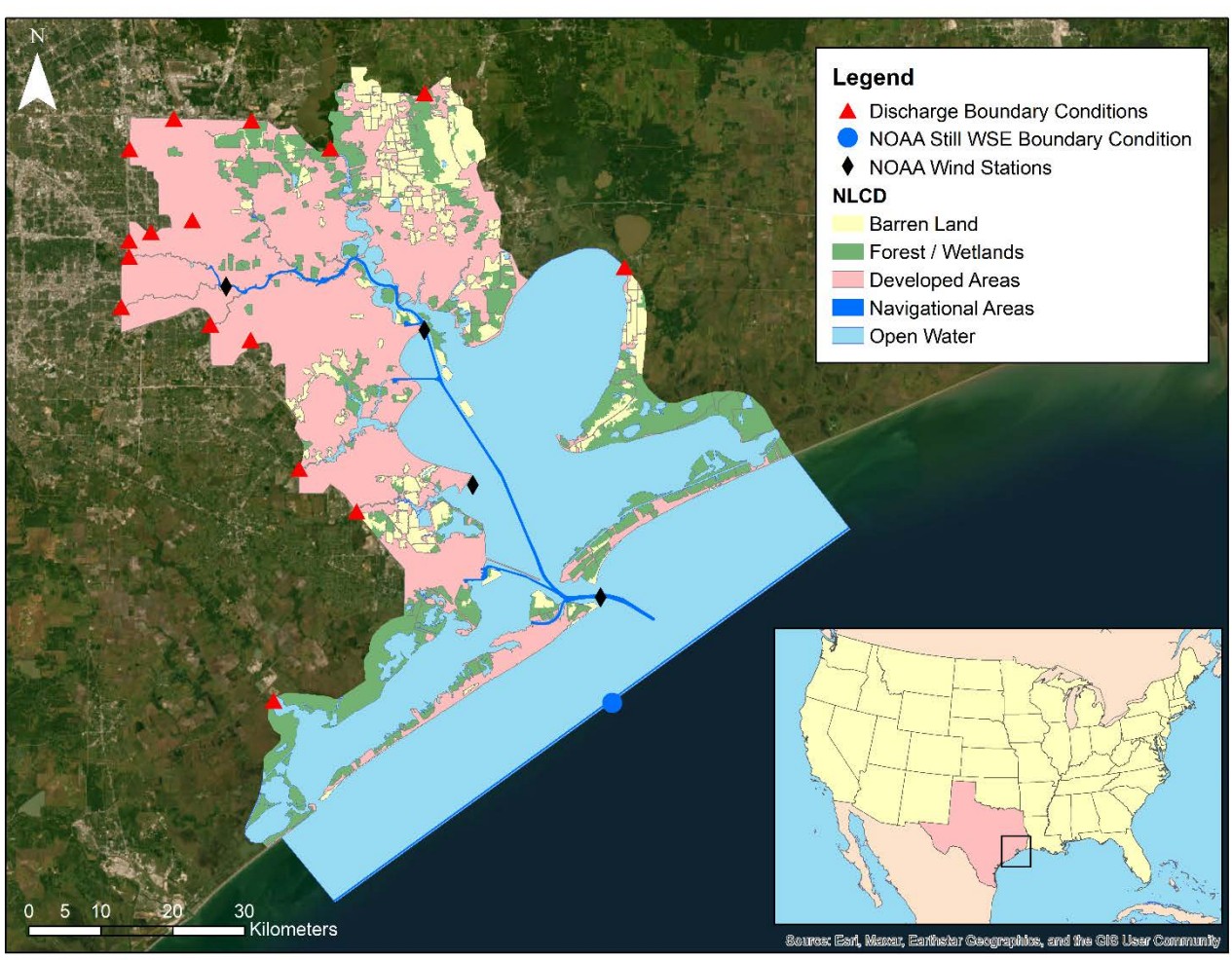


**Figure 2. Study area map with discharge, still water surface elevation and wind stations as boundary conditions. NLCD land covers are incorporated as manning's roughness in the HEC-RAS 2D model. Basemap ESRI World Imagery.**

### 3.2 Discharge and tidal forcings

Hourly discharge data from the U.S. Geological Survey (2016) is used for most of the streams incorporated within the HR2D
model. Missing data for some gauges are estimated by considering their correlation with other gages located upstream. The U.S. Army Engineer Research and Development Center (ERDC) has provided the daily discharge time series data for Dickinson Bayou, Chocolate River, and Trinity River. San Jacinto River discharge values are estimated using gage height time series from USGS gage Lk Houston nr Sheldon, TX (08072000). As downstream boundary condition, the hourly Stillwater elevation data from National Oceanic and Atmospheric Administration (NOAA) Galveston Bay Entrance station is selected.
Table 1 summarizes the boundary conditions applied to the HR2D model.

**Table 1. Summary of discharge and still water surface elevation boundary conditions used in the model setup.**

| Gauge station name | Source | Code/ID | Use |
|---|---|---|---|
| Galveston Bay Entrance | NOAA | 8771341 | Still water surface elevation downstream |
| Sims Bayou at Houston | USGS | 08075500 | Discharge, data estimated with values using USGS gauge 08075400 |
| Brays Bayou at Houston | USGS | 08075000 | Discharge |
| Buffalo Bayou at Houston | USGS | 08074000 | Discharge |
| Whiteoak Bayou at Houston | USGS | 08074500 | Discharge |
| Greens Bayou nr Houston | USGS | 08075900 | Discharge |
| Garners Bayou nr Humble | USGS | 08076180 | Discharge |
| Berry Bayou at Nevada | USGS | 08075605 | Discharge |
| Little Whiteoak Bayou at Trimble St | USGS | 08074540 | Discharge |
| Clear Ck nr Friendswood | USGS | 08077600 | Discharge |
| San Jacinto River nr Sheldon | USGS | 08072050 | Discharge, data estimated with height values over weir using USGS gauge 08072000 |
| Cedar Bayou nr Crosby | USGS | 08067500 | Discharge |
| Halls Bayou | USGS | 08076500 | Discharge |
| Hunting Bayou | USGS | 08075763 | Discharge |
| Goose Ck nr Mcnair | USGS | 08067520 | Discharge |

## 3.3 Precipitation and wind forcings

Extensive efforts have been dedicated to the detailed comparison and evaluation of diverse precipitation datasets generated on a regional or global scale. Within this framework, researchers have rigorously examined the total precipitation outputs derived from various sources, their alignment with alternative datasets, and their consistency with gauge-based measurements. The investigation into the spatial and temporal patterns of extreme precipitation events, particularly during Hurricane Harvey has become essential due to the event's catastrophic impact (Fagnant et al., 2020; Wang et al., 2018). Researchers have taken

a comprehensive approach, encompassing a broad spectrum of precipitation products, which include both remote sensing and model-based estimations. The comparison often extends to not only the total accumulated precipitation but also its spatiotemporal distribution, intensity, and duration. This multifaceted evaluation aims to discern the differences in performance, uncover potential biases, and ascertain the overall reliability of these estimates (Brauer et al., 2020; Chen et al., 2020; Gao et al., 2021; Habibi et al., 2021; Omranian et al., 2018).

In this study, an evaluation of seven distinct precipitation products is made, across the temporal and spatial resolutions that are conducive to capturing the intricacies of hydraulic routing through HR2D. The precipitation products considered for Cop-BMA assessment include:

1.  CMORPH (Climate Prediction Center MORPHing technique) (Xie et al., 2019)

2. Daymet (Daily Surface Weather Data on a 1-km Grid) (Thornton et al., 2022)


3. ERA5 (Muñoz Sabater, 2019)

4. IMERG (Integrated Multi-satellitE Retrievals for GPM) (Huffman et al., 2019)

5. Multi-Radar Multi-Sensor (MRMS) (Zhang et al., 2016)

6. NCEP Stage IV precipitation data (Du, 2011)

7. NLDAS-2 (North American Land Data Assimilation System version 2) (Xia et al., 2009)

To facilitate analysis and modelling, these datasets undergo preprocessing in the Hydrologic Modelling System (HEC-HMS) software to generate .dss files, thus facilitating their integration into the HEC-RAS Unsteady Flow Meteorological Data.

It is important to emphasize that while the primary focus of this research is to assess the integration of precipitation data in compound flood events, certain limitations exist. Notably, NLDAS and Daymet products do not provide coverage for terrain areas near the coastline, particularly in the southern region of the model domain, which includes Galveston and Texas City.

This geographical limitation underscores the need for careful consideration when interpreting and generalizing the findings within these specific regions.

In addition to the seven precipitation products mentioned above, rain gauge data (RG), provided by the Harris County Flood Warning System (HCFWS) portal (https://www.harriscountyfws.org/) is integrated into the study as comparison for modelling results. The Houston metropolitan region comprises a network of 188-gauge stations distributed across the county. For this

study, a subset of 20 stations is selected within the study domain, ensuring the availability of continuous rainfall data specifically during the occurrence of Hurricane Harvey over the city of Houston. To facilitate the integration of these rain gauge measurements as a spatially distributed data, the Inverse Distance Squared Weighting (IDW) interpolation method is employed (Chen and Liu, 2012). This technique allows for the estimation of precipitation values at locations that do not have direct measurements by considering the spatial proximity and inverse distances between available gauge stations.

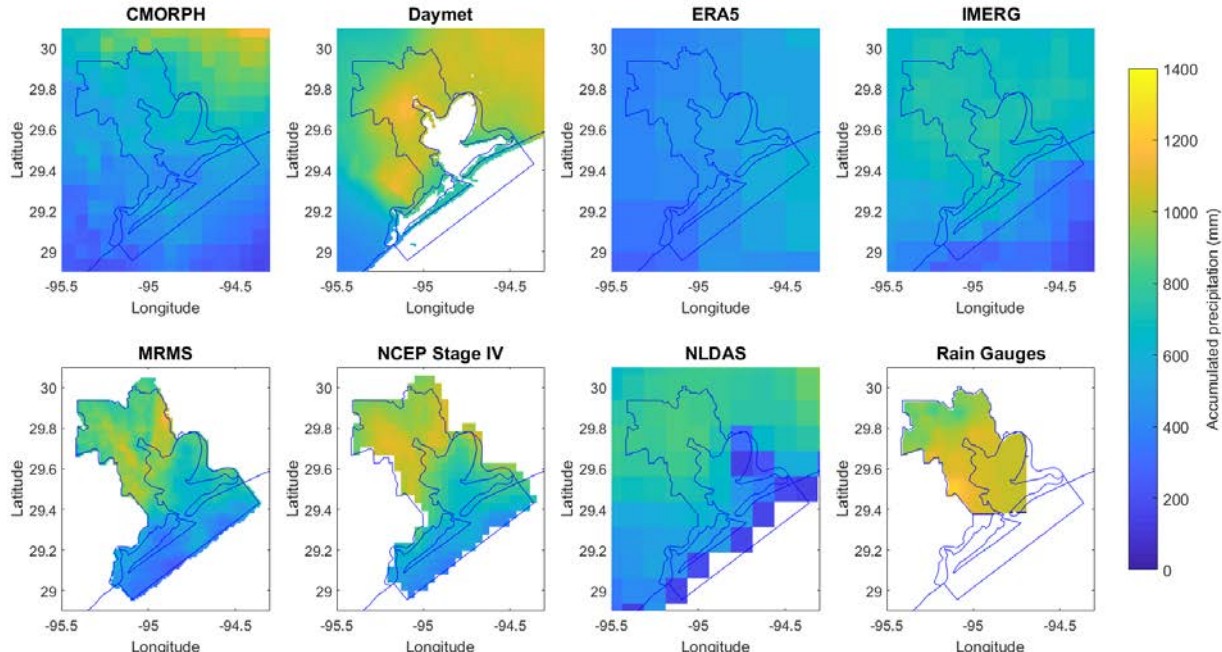

**Figure 3. Spatial distribution accumulated precipitation of the seven different precipitation datasets, rain gauges data, and their coverage over the study area during Harvey event from 23/Aug/2017 to 03/Sep/2017.**

**Table 2. Spatial and temporal details of the eight precipitation products used in this study.**

| Precipitation product | Spatial resolution (Approx.) | Temporal resolution | Observations |
|---|---|---|---|
| NLDAS-2 | 12.5km x 12.5km | Hourly | -Do not cover coastal domain |
| Daymet | 1km x 1km | Daily | -Do not cover coastal domain |
| CMORPH | 7.77km x 7.77km | 30 min | |
| IMERG | 11.1km x 11.1km | 30 min | |
| ERA5 | 31km x 31km | Hourly | |
| MRMS | 1km x 1km | Hourly | |
| NCEP Stage IV | 4.76km x 4.76km | Hourly | |
| Rain Gauges | 1km x 1km | 15 min | -Do not cover coastal domain *Rain interpolated between 20 rain gauges within HEC-RAS |

Hurricane Harvey made a significant impact on the Galveston Bay region, manifesting itself as a tropical storm characterized by varying maximum wind speeds. These speeds ranged from 78.5 km/h to 34.6 km/h, spanning from the entrance of Galveston to downtown Houston. Given the considerable length of the Galveston estuary, incorporating wind forcing into the study is essential to comprehensively account for its hydrodynamic behaviour over the surface of the water. Hourly wind velocity and

direction data were integrated from specific NOAA stations across the study area. These stations include Galveston Bay Entrance (8771341), Eagle Point (8771013), Morgans Point (8770613), and Manchester (8770777). These meteorological boundary conditions are utilized into the HR2D model to accurately simulate the effects of wind within the hydrodynamic system. Lagrangian reference frame and Andreas et al. (2012) drag formulation are selected. Similar to precipitation data, IDW method is also selected for wind spatial interpolation along the study area.


## 4. Results and Discussion

The simulations conducted within the HR2D model involved fixed Manning coefficients, ensuring that the water surface elevation is solely influenced by the applied precipitation forcing. A model warm-up period is set from August 16, 2017, to August 23, 2017. The results during this interval are exclusively used to calibrate the roughness coefficients in comparison to
observational data. The comprehensive assessment of the model's performance is conducted over the period from August 23, 2017, to September 3, 2017, with hourly results. This temporal scope encompasses the passage of Hurricane Harvey and the subsequent recession of the water levels.

Figure 4 presents hourly hydrographs of observed water surface elevation (WSE) data alongside simulated outputs for various validation stations (Information of validation stations is in Table S1 in supplementary data). The simulation results highlight
that relying on a single QPE does not lead to consistent responses across the evaluated hydrographs. It becomes apparent that some stations experience an overestimation of water levels, while in other areas within the region, the response tends towards underestimation for the same product. A notable case is observed with the Daymet product, which has a finer spatial resolution (1km x 1km), yet its daily precipitation values struggle to capture the hourly fluctuations evident in the observed data. Notably, the hydrograph results derived from the Daymet product exhibit a step-like behaviour on a daily scale in several validation
stations, particularly within the upper reaches of the modelled watersheds.

Among several validation stations, a discernible alignment between observed values and ensembles generated by different precipitation datasets can be observed. However, it is crucial to acknowledge that in certain instances, the variability among ensembles can exceed 2 meters across different products, and certain ensembles fail to accurately replicate the behaviour of observed values. These observations underscore the challenges involved in accurately reproducing the temporal and spatial
patterns of precipitation, especially in regions characterized by complex topography and intricate watershed characteristics and influenced by structural uncertainty or parametrization within HR2D model. Additionally, inland initial infiltration processes that might have occurred during the Hurricane Harvey event could have impacted the results of water surface elevation at gauges in watersheds modelled and were not considered in the hydrodynamic model. Furthermore, in highly urbanized systems, drainage systems play a significant role during storm events. Due to the limitations of the employed model,
such hydrosystems are not included in the simulations, adding a layer of uncertainty due to the model structure and the type of physical processes involved.

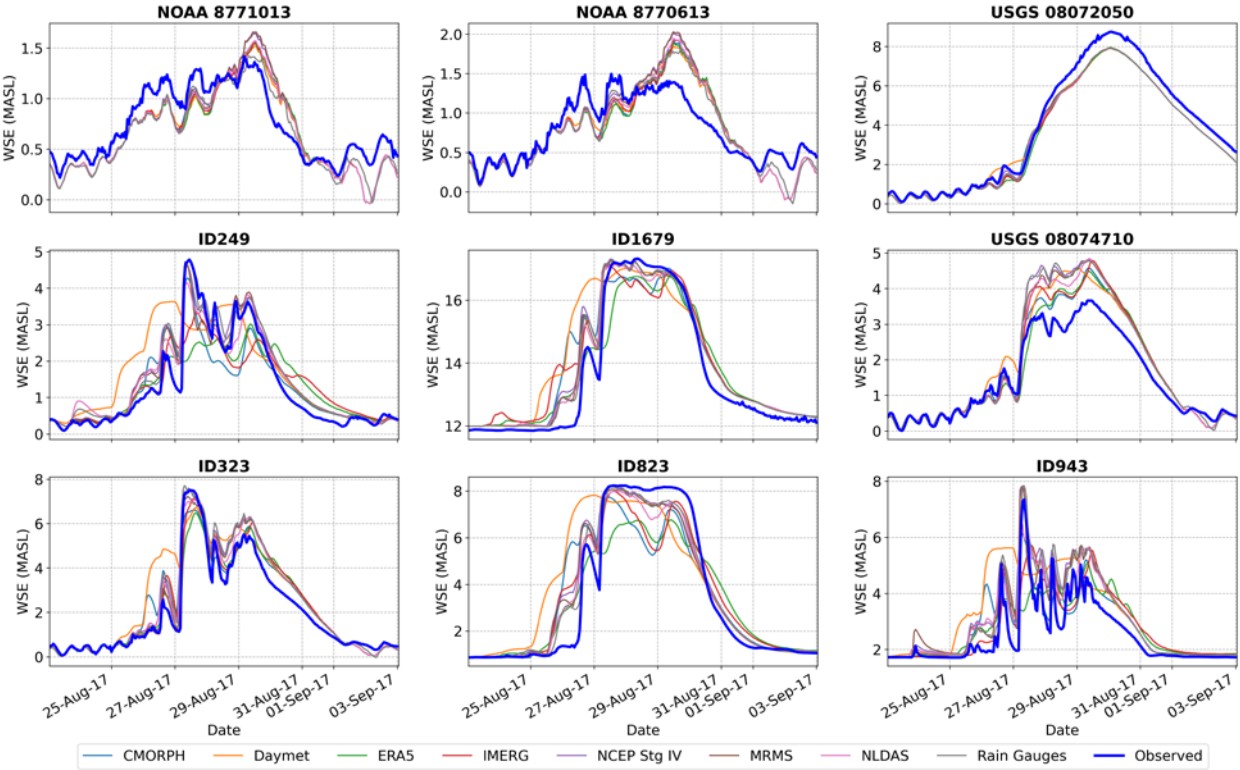

**Figure 4. Hydrographs of simulated water surface elevation (WSE) by the HEC-RAS 2D model using eight different precipitation datasets along with the observed WSE values observed for Hurricane Harvey. Each subplot represents the result at different validation stations where ID refers to stations in the Harris County Flood Warning System.**

### 4.1 Global Cop-BMA flood elevation and mapping extent

With the integration of the Cop-BMA approach, it becomes feasible to enhance the accuracy of flood depth estimates at each validation station. Nonetheless, the generation of results while considering their spatial distribution along a large domain can be streamlined through clustering techniques.

For this purpose, the K-means method is used to partition the 30 validation stations along the study area from different sources (USGS, NOAA and HCFWS) into three primary clusters, a selection determined by applying the Elbow method to identify the optimal K value. Clustering is implemented by utilizing a flood range metric, defined as the difference between the peak value and the initial observed value at the beginning of the Hurricane Harvey evaluation period. In this method, each validation station is associated with an area of influence, which is delineated based on topographic attributes and often coincides with watershed concentration points. In some instances, engineering expertise is employed to supplement the delineation process.

Figure 5 shows the spatial configuration of validation stations, their corresponding areas of influence, and the resultant clustering regions within the study area. This strategic clustering allows for a more focused and structured analysis, facilitating

the extraction of meaningful insights from the ensemble data generated by different precipitation products.

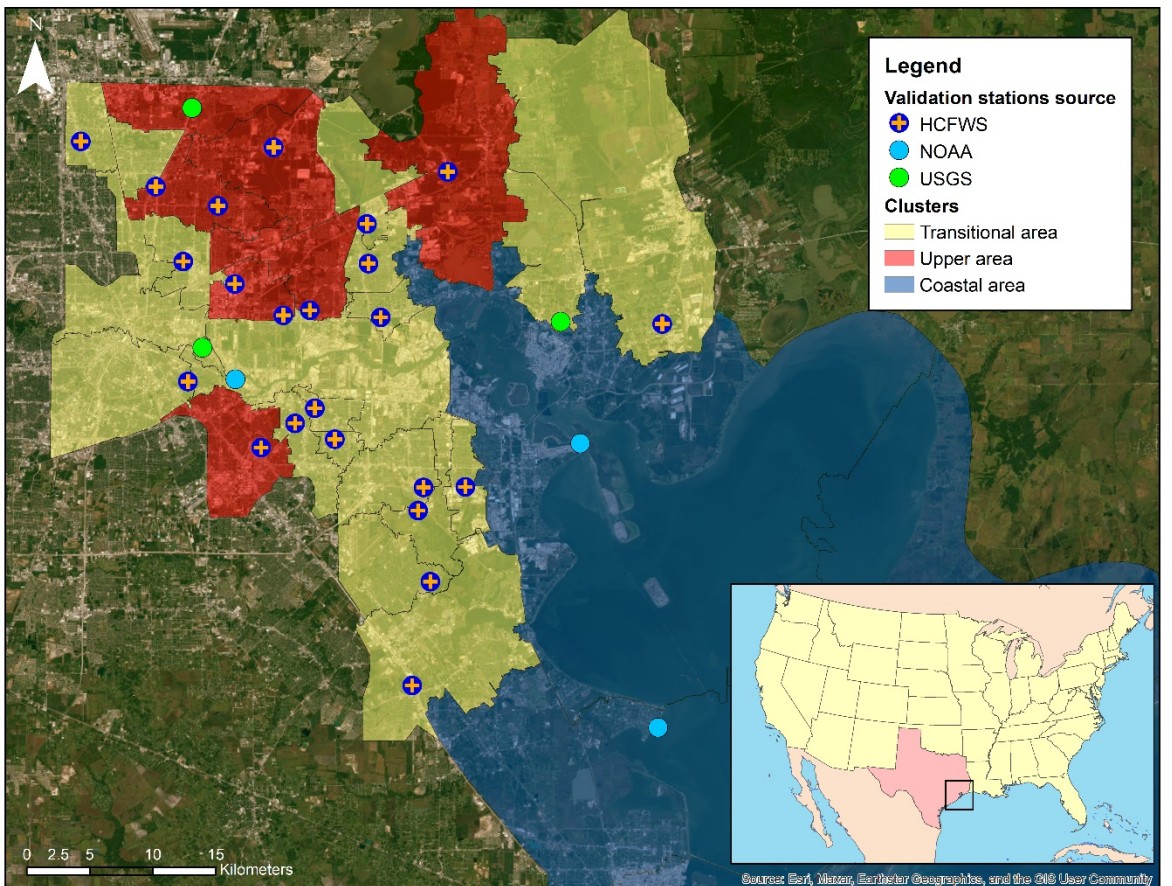

**Figure 5. Location of validation stations, areas of influence and clustering regions in the study area. The entire domain is clustered into three regions of coastal, transitional, and upper areas. Basemap ESRI World Imagery.**


A crucial step in implementing Global Cop-BMA is to fit marginal distributions of observed and simulated data and determine the copula parameters that define the underlying correlation structure of the multivariate distribution. To fit marginal distributions, an array of probability distributions undergo testing. This comprehensive evaluation includes a variety of distributions such as Cauchy, Gumbel, Alpha, Beta, Gaussian, Exponential, Gamma, Lognormal, Generalized Pareto,

Generalized Extreme, Weibull, and others. Given the intrinsic nature of the data in this study, which comprises water surface

elevation data in coastal environments, it is essential to choose statistical distributions that accommodate both positive and negative values within their range of support. Parameter estimation for each distribution is performed using the Maximum Likelihood Estimation (MLE) technique. To identify the most suitable marginal distribution, the sum of squared errors (SSE) is employed to facilitate the selection process, choosing the distribution that provide the lower SSE value.

Table 3 provides a summary of the optimal fits of marginal distributions for various outcomes of the hydrodynamic modelling. The outcomes are categorized by each precipitation product and grouped according to their respective clusters. The table also includes the estimated value of SSE between the empirical CDF and the fitted CDF values.

**Table 3. Summary of marginal distribution fitting results per precipitation product and sum of squared errors for the**
**best distribution.**

| Precipitation Product | Transitional cluster | | Upper cluster | | Coastal cluster | |
|---|---|---|---|---|---|---|
| | Best marginal | SSE (m) | Best marginal | SSE (m) | Best marginal | SSE (m) |
| CMORPH | Pearson type3 | 1.236 | Beta | 2.104 | Beta | 0.827 |
| Daymet | Exponential | 2.115 | Beta | 2.201 | Beta | 0.886 |
| ERA5 | Genpareto | 0.919 | Beta | 2.738 | Beta | 0.744 |
| IMERG | Genpareto | 1.415 | Beta | 2.822 | Beta | 2.534 |
| NCEP Stage IV | Gamma | 2.559 | Beta | 3.274 | Gamma | 0.775 |
| MRMS | Pearson type3 | 2.111 | Beta | 3.073 | Pearson type3 | 0.730 |
| NLDAS | Gamma | 1.936 | Beta | 3.089 | Beta | 0.768 |
| Observed data | Pearson type3 | 3.325 | Beta | 3.853 | Gamma | 1.457 |

Upon identifying the optimal marginal distributions, the subsequent stage of the Global Cop-BMA framework involves the selection of a copula function. This copula function serves as a vital link, effectively connecting the CDFs of model simulations with observed data. Among various copula options, the most pertinent selection is the one that efficiently captures the inherent

dependence structure between the variables being analysed. In this study, five distinct copula functions are evaluated, Gumbel, Clayton, and Frank from the class of Archimedean copulas, and Gaussian and t-Student from Elliptical group. Copulas are constructed and evaluated using the marginals distributions of the observed data and each of the precipitation products modelling results of water surface elevation per cluster as $c(u_y, u_{M_k})$. Fitting and selection process was conducted using Akaike Information Criterion (AIC) and copula cross-validation criterion (xv-CIC) (Grønneberg and Hjort, 2014) using copula

package implemented in R (Hofert et al., 2023), where the copula fit with lowest value of AIC and higher xv-CIC was selected. Table 4 shows the selected copulas for the seven QPEs evaluated in HR2D simulations over the three clusters. Calculated values for AIC and xv-CIC are presented in Table S2 in the supplementary material.

**Table 4. Summary of copula fitting results per cluster for each precipitation product used in the HEC-RAS 2D model simulations.**

| Precipitation Product | Transitional cluster | Upper cluster | Coastal cluster |
|---|---|---|---|

| CMORPH | Gumbel | Gumbel | t-Student |
|---|---|---|---|
| Daymet | Gumbel | Gumbel | t-Student |
| ERA5 | Gumbel | Gaussian | t-Student |
| IMERG | Gumbel | Gumbel | t-Student |
| NCEP Stage IV | Gumbel | Gumbel | t-Student |
| MRMS | Gumbel | Gumbel | t-Student |
| NLDAS | Gumbel | Gumbel | t-Student |


After applying the EM algorithm, it becomes feasible to compute the hydrograph generated for each station based on the estimated weights for each cluster. The averaged error of simulations using different QPEs against observations from validation stations are shown in Figure 6, featuring rain gauges simulation errors and estimations for Global Cop-BMA approach per cluster. Notably, this method exhibits better results in its responses to different precipitation products and clusters, leading to

an enhanced accuracy in water level estimations, particularly during peak periods compared to the range of modelling water surface elevation outputs from the analysed QPE such as Daymet or ERA5 which exhibit larger averaged errors. This demonstrates the Cop-BMA's capability to generate results that closely correspond to the observed values at the validation stations. It is important to highlight that if all models consistently overestimate or underestimate, Global Cop-BMA may not lead to significant improvement in the result (e.g., NOAA 8770613, USGS 08074710, USGS 08072050 in Figure 4; and

Coastal cluster in Figure 6). Despite its advanced weighting mechanism, Global Cop-BMA's effectiveness relies on the diversity and accuracy of the model ensemble. Therefore, while it enhances the integration of diverse model outputs, its capability to improve results may be limited when all models exhibit similar differences compared to the observations at certain sections of the hydrograph. The process of selecting validation stations within each cluster holds a significant influence over the subsequent calculation of weights using the BMA methods. The choice of metric or clustering technique can yield distinct

combinations of validation stations, subsequently leading to varying weight distributions.

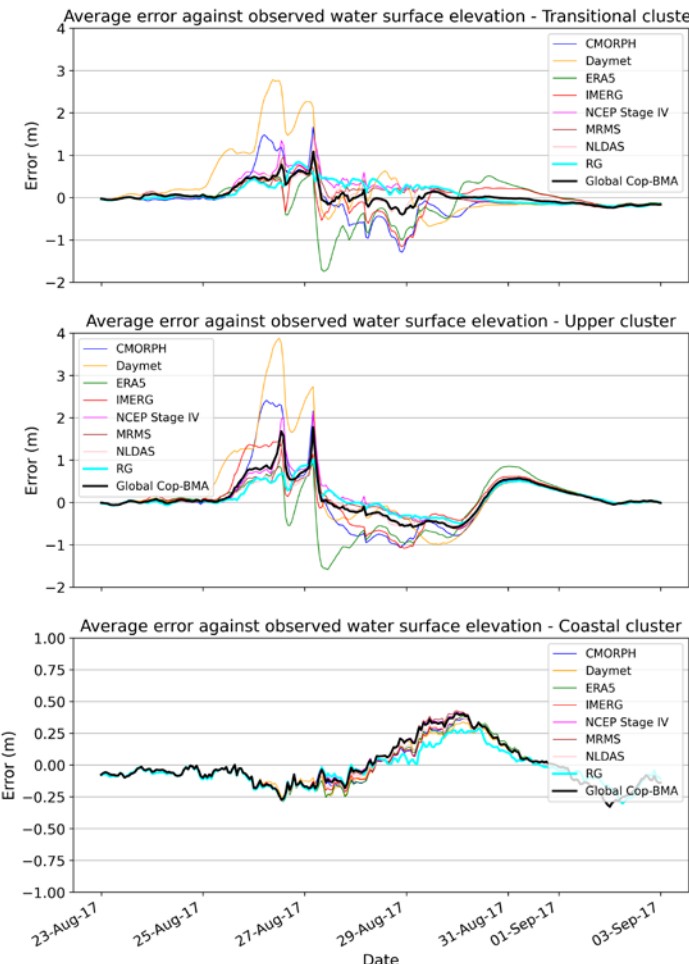

**Figure 6. Averaged error time series of validation stations per cluster of simulated water surface elevation (WSE) by the HEC-RAS 2D model using QPE datasets and Global Cop-BMA approach results (black) against observed WSE during Hurricane Harvey.**

Figure 7 shows the calculated weights for the Global Cop-BMA method across the three analysed clusters. The weights show the contributions of each QPE within different clusters. The distinct distribution of weights between the three clusters reflects their unique strategies in handling uncertainties and variations among different precipitation products.

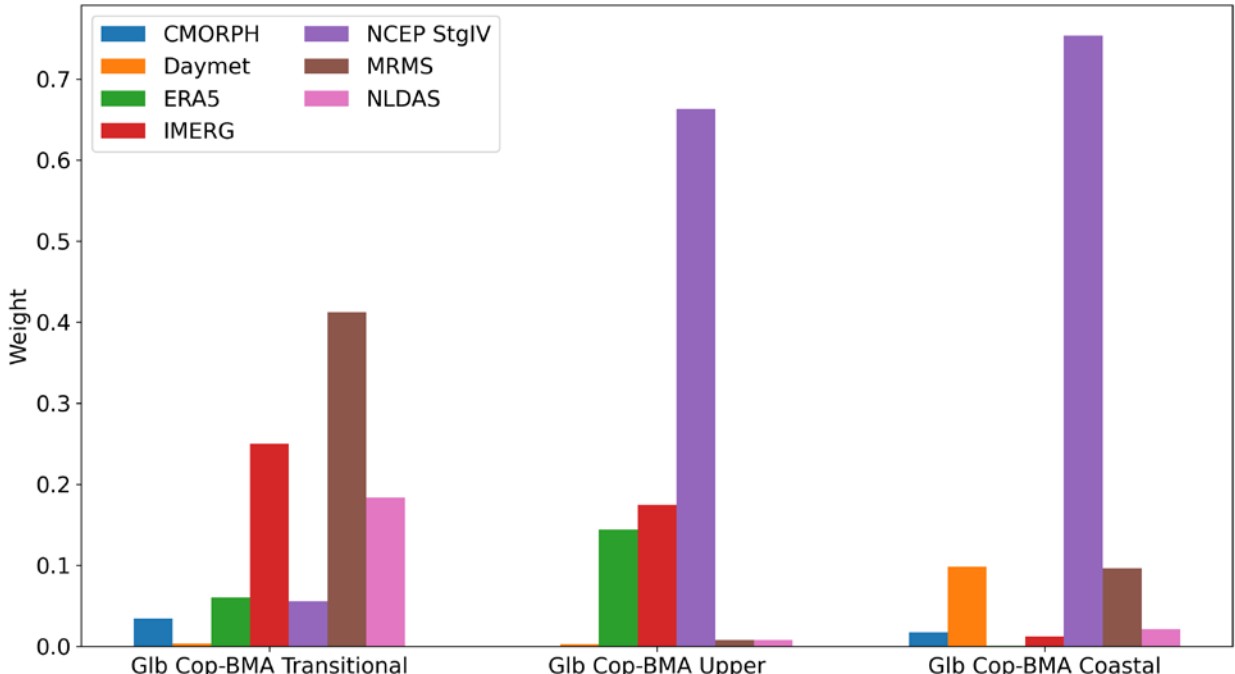

**Figure 7. Summary of calculated weights of the different precipitation products used within the Global Cop-BMA approach. The weight calculation task is implemented in each cluster separately.**

As depicted in Figure 7, the distribution of weights per cluster exhibits greater variability. In the transitional cluster, CMORPH, Daymet, ERA5 and NCEP Stage IV have weights below 0.1, as these four products generated underestimated responses in the hydrographs for most stations within this cluster. Weights center around the precipitation from MRMS, IMERG and NLDAS QPEs. Within the upper cluster, a different weights distribution among the QPEs is observable, with minimal influence from CMORPH, Daymet, MRMS and NLDAS QPEs. A higher difference is observable the three more dominant QPE, where NCEP Stage IV has a weight of 0.663 compared to the 0.144 of IMERG and 0.174 of ERA5 QPEs. For the Coastal cluster, precipitation from Stage IV QPE also holds the greatest weight (0.753) compared to the rest of QPEs which hold weight values below 0.1.Within this cluster, minimal discernible differences exist between QPEs water surface elevation results for the stations, as seen in Figure 4 (NOAA stations 8771013 and 8770613) and in Figure 6.

The evaluation of model performance in validation stations is measured through different metrics, including Nash-Sutcliffe Efficiency (NSE) (Nash and Sutcliffe, 1970), Kling-Gupta Efficiency (KGE) (Kling et al., 2012), Root Mean Square Error (RMSE), and Mean Bias Error (MSE). The formulations of these metrics, which collectively provide insights into different facets of model accuracy, are summarized in Table 4. These metrics serve as quantitative measures to assess the model's capability in capturing the observed variations in water surface elevation during the Hurricane Harvey event and subsequent recession phase.

**Table 5. Summary of four main performance metrics used in this study for validating predicted time series of WSE compared to observed values.**

| Evaluation metric | Equation |
|---|---|
| Root Mean Square Error | $RMSE = \sqrt{\dfrac{\sum_{i=1}^{N}(y_i - \widehat{y_i})^2}{N}}$ |
| Mean Bias Error | $MBE = \dfrac{1}{N}\sum_{i=1}^{N}(\widehat{y_i} - y_i)$ |
| Nash-Sutcliffe Efficiency | $NSE = 1 - \dfrac{\sum_{i=1}^{N}(\widehat{y_i} - y_i)^2}{\sum_{i=1}^{N}(y_i - \bar{y})^2}$ |
| Kling-Gupta efficiency | $KGE = 1 - \left\{ \left[\dfrac{cov(y,\widehat{y_s})}{\sigma_o \sigma_s} - 1\right]^2 + \left[\left(\dfrac{\sigma_s}{\sigma_o}\right) - 1\right]^2 + \left[\left(\dfrac{\bar{y_s}}{\bar{y}}\right) - 1\right]^2 \right\}^{\frac{1}{2}}$ |

$N$: total time steps, $i$: time step, $y_i$: observed data, $\bar{y}$: mean of observed data, $\widehat{y_i}$: model simulation, $\bar{y_s}$: mean of model simulations, $\sigma_0$: standard deviation of observed data, $\sigma_s$: standard deviation of model simulations.

Figure 8 provides a comprehensive overview of collective performance metrics of the HR2D model across the seven QPE simulations, rain gauges simulation, and the Global Cop-BMA multi-modelling for the seven QPEs evaluated at 30 validation stations over the 11-day simulation period. In general, the inundation modelling driven by different products consistently exhibits NSE performance with mean values ranging between 0.695 and 0.846 In terms of KGE performance, the interquartile ranges for QPEs display broader ranges, and the medians for Daymet and ERA5 products fall below 0.8 in contrast to other simulations.

Notably, the Cop-BMA approach exhibits slightly higher performance metrics compared to the QPE products, NSE has an average of 0.858 and its total variability is lower compared to single precipitation products. KGE metric has a similar result with an average value of 0.852. The Averaged RMSE for Cop-BMA is 0.561m which is smaller than all the single QPE except for the rain gauges simulation which is only 3 centimetres lower. The averaged MBE for single QPEs ranged between -0.018 and 0.23m, while the Global Cop-BMA method results in an averaged value of 0.049m. Among individual products, the rain gauge outperforms all spatially distributed precipitation datasets and comes closest to matching the performance of Cop-BMA method. This highlights that reanalysis gridded precipitation products may have higher errors when compared to in-situ rain observations, and allows Global Cop-BMA to generate QPE post-processed results that are closer to modelling results with observed precipitation from rain gauges. This methodology could be replicated in areas where measured precipitation is not available and obtain better performance metrics accounting for the uncertainties from this input. Another factor is that our study area encompasses only a few grid cells of some reanalysis products, making the advantages of using spatially distributed data less apparent. Overall, the global Cop-BMA approach offers two advantages over individual products: first, it improves and diminishes the variability of performance metrics over different locations, underscoring the robustness of the proposed

approach. Second, it accounts for uncertainties associated with individual precipitation products and generates probabilistic flood inundation maps as a post-processing methodology.

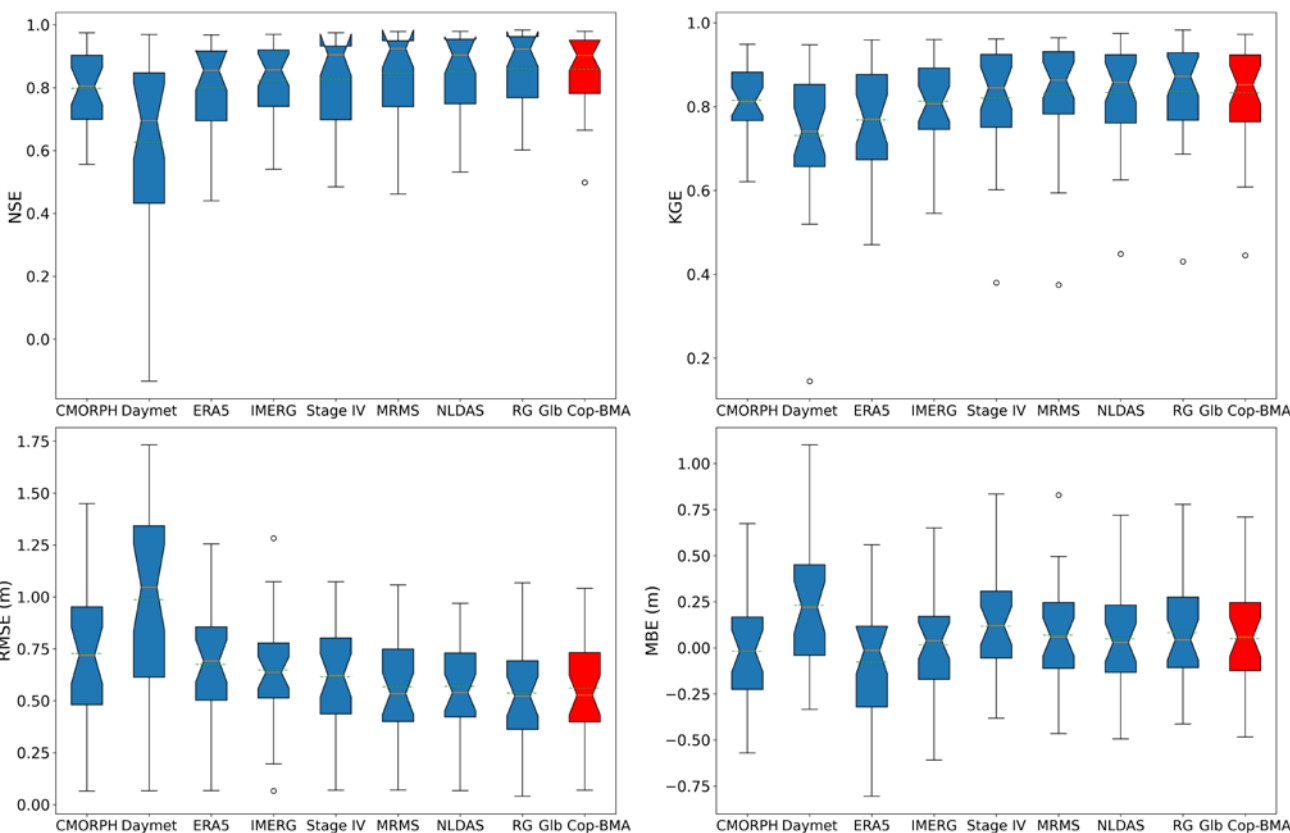

Figure 8. Boxplot of four performance metrics for different precipitation products (blue) as well as Global Cop-BMA results (red). The boxes represent the distribution of performance metrics across the validation stations.

Utilizing the defined areas of influence and the established clusters, a crucial step for probabilistic flood inundation mapping involves the creation of a mask that applies the calculated weights of each QPE and rain gauge product. The resulting water depth simulations from the HR2D model are then exported in raster format. Employing raster calculator functions, the probability of flooding can be quantified using binary flood raster maps. In these maps, pixels hold a value of 0 to denote the absence of water and 1 if water is present. Figure 9 presents the computed flood depth and the corresponding estimated flood probability using the weights calculated with the Global Cop-BMA method for the modelled area close to downtown Houston. This approach offers a probabilistic understanding of the potential flooding scenario, providing decision-makers and stakeholders with valuable insights into the likely extent and severity of flooding.

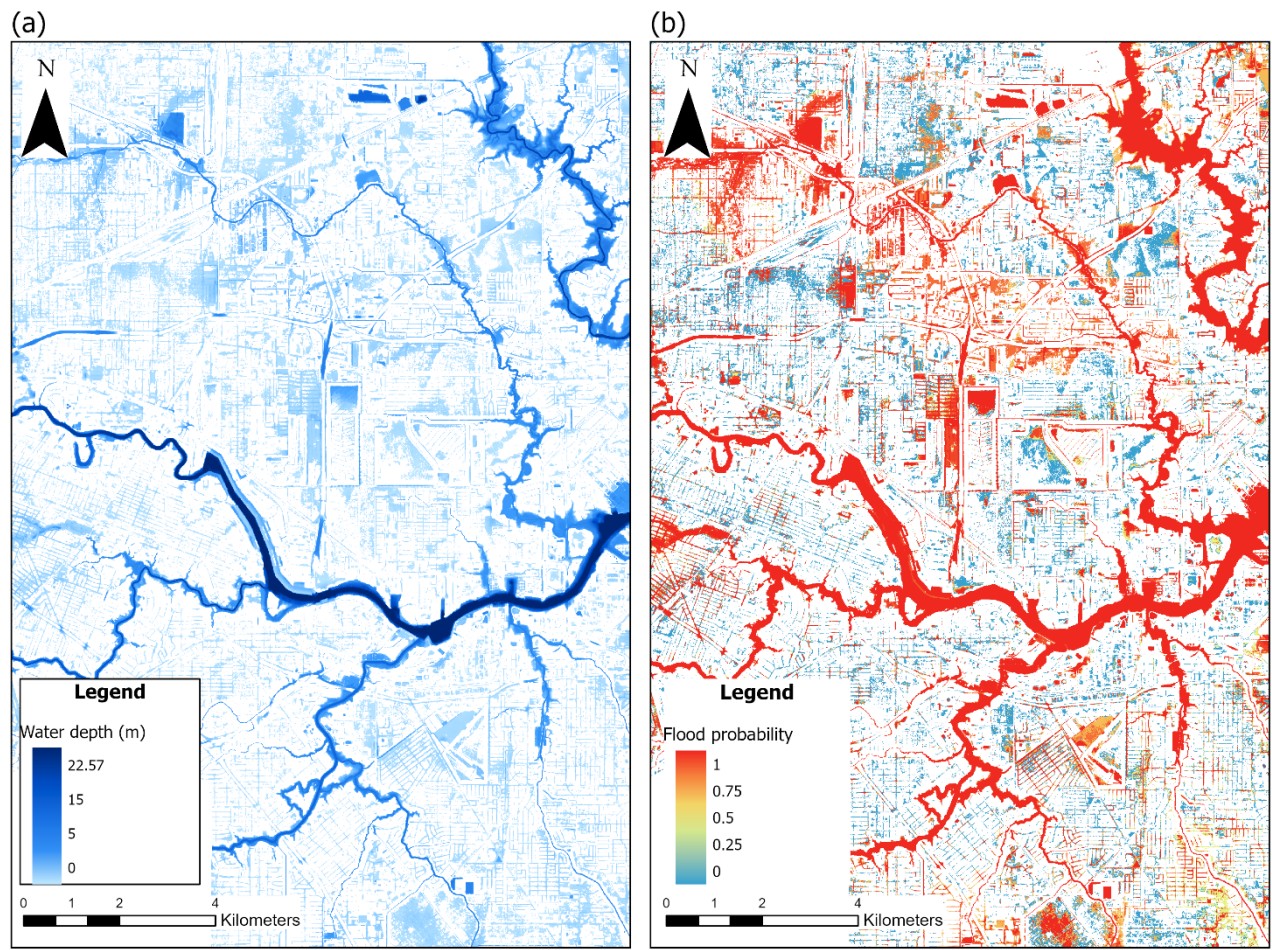

**Figure 9. Results of probabilistic flood inundation map using Global Cop-BMA methodology for Hurricane Harvey event over Houston area. (a) The average of water depth maps generated by Global Cop-BMA approach (b) The probabilistic flood extent map provided by Global Cop-BMA.**

Given the limited availability of satellite images and validation information for the complete extent of flooding, the challenge lies in generating accurate spatial information for validation purposes over Houston (Saksena et al., 2020). While the presented approach offers a robust method for probabilistic flood inundation mapping, the verification of spatial extent remains a crucial task. The validation tasks were primarily focused on assessing the performance of model outputs at validation stations, as depicted in Figure 5. This approach enabled us to calculate the performance metrics of WSE over a well-distributed network of stations with remarkable temporal resolution. Data collected from these validation stations sufficiently capture the hydrograph behaviour within the study domain and enables us to quantify flood extents in a probabilistic manner using the

HR2D model incorporated with the Cop-BMA method. It is worth noting that while a flood inundation map provided by a single QPE may potentially exhibit greater accuracy compared to one generated by Cop-BMA, the primary advantage of using Cop-BMA lies in its ability to generate probabilistic flood inundation maps while considering uncertainties associated with various QPE sources. Additionally, the QPE offering the highest accuracy is not consistently a single product; it may vary across different study cases and flood event characteristics. Therefore, employing a BMA-based approach could be a viable strategy to achieve high accuracy while accounting for of uncertainties. Future research efforts may focus on improving the validation process in other study areas through the integration of additional data sources and innovative techniques to validate the entire extent of flooding accurately with other sources more than gauging stations and high water marks, especially in highly urban environments with rapid urbanization and constant land cover changes, and also over large and with high resolution computational domain (Juan et al., 2020; Schubert et al., 2022).

## 5. Discussions and Conclusions

Dynamic simulation of extreme flood events demands a comprehensive approach that accounts for the inherent uncertainties and limitations present in both forcing data and numerical models. When conducting scenario analysis by inundation modelling driven by different precipitation forcings across the domain, it is crucial to acknowledge that definitively asserting the superiority of one product over another is not feasible. This is due to their inherent limitations in terms of spatial and temporal coverage, as well as the estimated precipitation values given the algorithms or methodologies used to generate the QPEs. In this study, comprehensive validation was feasible due to the access to a dense network of stations over Harris County of in-situ precipitation data (rain gauges) and water surface elevation with temporal high resolution. However, such data are not widely available in many regions at a comparable density and temporal resolution. The substantial variability in the modelling results, both in terms of flood extent and water depth, is evident, leading to instances of both overestimation and underestimation throughout the response hydrograph for all assessments conducted by the different precipitations inputs as forcing to HR2D model.

The utilization of Bayesian Model Averaging tools operates on the premise that there is not a single best model, specifically a precipitation product that fully captures the behaviour of the flooding caused by Hurricane Harvey. Similarly, there is not a single BMA scheme that universally outperforms any other approximation (Parrish et al., 2012). It has been shown that the assumption of data and conditional PDF follows a Gaussian distribution, as imposed by the BMA approach in many hydrologic applications, may lead to an oversimplification of extreme event behaviour, affecting the calculated weights and subsequent flood predictions. In this regard, it has been suggested that the incorporation of copula functions (Cop-BMA) can enhance the characterization of model-dependence generated by hydrodynamic water surface elevation data distributions and their relationships with observed data. Results using Cop-BMA approach show better distribution of performance variability metrics over the validation stations and reduced the averaged error per cluster compared to single QPEs in the evaluated metrics.

Given the sensitivity of weight distributions to the selection of validation stations and clustering techniques, future studies could explore the impact of alternative clustering methods or metrics on the overall outcomes of the Global Cop-BMA approach. Such investigations could provide insights into the robustness of the method and its ability to adapt to varying configurations of validation data. Understanding how different clustering strategies influence weight distributions will contribute to a comprehensive interpretation of the uncertainty associated with flood predictions and further refine the decision-making process in flood risk management.

One advantage of our proposed framework is its flexibility allowing for the use of alternative precipitation products to enhance model simulations. For instance, this framework can be implemented for operational forecasting purposes where the Quantitative Precipitation Estimations (QPEs) utilized in this study can be replaced with Quantitative Precipitation Forecasts (QPF) from numerical weather prediction models such as High-Resolution Rapid Refresh (HRRR), North American Mesoscale Forecast System (NAM), Global Forecast System (GFS), European Centre for Medium-Range Weather Forecasts (ECMWF) among others. Additionally, the proposed framework can be further improved by accounting for uncertainties related to various factors such as boundary conditions, and Digital Elevation Models (DEMs), which have already been analysed separately and individually. The HEC-RAS model can also incorporate the impact of infiltration during flood events. This involves testing various infiltration methods, such as Deficit and Constant, SCS Curve Number, and Grenn-Ampt, across different storm events in rural areas with diver land cover. By considering these additional sources of uncertainty within the modelling process, it is possible to enhance the accuracy and reliability of probabilistic flood inundation mapping, providing a more holistic perspective on extreme event simulations. This approach would yield a deeper understanding of the complex interactions and non-linearity of multiple factors contributing to flood events, thereby contributing to more robust flood risk assessments and management strategies. The challenge of scarce validation data for flood extents was addressed by generating probabilistic inundation maps. These maps assist in decision-making, especially in coastal regions where risk assessment is particularly complex. However, further research is needed to validate these spatial estimates. This is especially relevant in coastal regions where the interplay of various forcings makes it particularly complex to estimate risk scenarios for specific return periods. One limitation of the employed numerical model is its inability to directly incorporate the drainage networks present in urban areas. While the assumption that the drainage system was operating at 100% capacity, future research could explore the influence of these systems on accurately estimating water depth in urban areas at the city scales. Additionally, considering infiltration processes in hydrodynamic modelling when driven by different precipitation products can improve flood inundation modelling skill.

**Data availability**

All the data used in this study, including the gauge discharge, water stage data and the DEMs are publicly available from the USGS, NOAA and Harris County Flood Warning System websites, respectively. All Precipitation data used in this study are

publicly available in their respective websites. ERDC has provided discharge data in Dickinson Bayou, Chocolate River, and Trinity River.

**Author contribution**

FG, KJ and HMd conceptualized the study. FG implemented the methodology, conducted formal analysis, generated results, and wrote the original draft. KJ provided guidance on the methodology, formal analysis and results and edited the original

draft. HMf – edited the original draft and suggested formal analysis. HMd – Participated in supervision, funding acquisition, suggested formal analysis and editing the original draft.

**Competing interests**

The authors declare that they have no conflict of interest.

**Acknowledgements**

This Study was financially supported by USACE contract # W912HZ202005.

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
