# Peer review of "Probabilistic Flood Inundation Mapping through Copula Bayesian Multi-Modelling of Precipitation Products"

_Natural Hazards and Earth System Sciences, 2024_

## Author Comment (AC1)

**Response to Referee 1 Comments on:**

**Probabilistic Flood Inundation Mapping through Copula Bayesian Multi-Modelling of Precipitation Products**

Francisco Javier Gomez, Keighobad Jafarzadegan, Hamed Moftakhari, and Hamid Moradkhani

MS. Ref. No.: NHESS-2024-26
Natural Hazards and Earth System Sciences

We appreciate the constructive comments provided by the reviewer as they have helped enhance the quality of our manuscript. Enclosed, please find the revised manuscript with all the necessary corrections implemented in accordance with the comments and suggestions in tracked and cleaned versions. The following are point-by-point answers, in blue color, to the comments of the first referee:

**Referee #1 (RC1):**

1. **The role of copula function and the choice for a specific function remains somewhat elusive.** For one, Section 2.2 could benefit from being more specific in the role of copula function in replacing the posterior distribution. How does the copula function relate to the prior distribution? In addition, the section would benefit from an introduction of the concept of a copula function in statistical terms so as to convince the reader that the separation of functional dependence and modeling of marginal distributions is appropriate.

   **Response:** We appreciate the reviewer's suggestion regarding the introduction of copula functions, which we consider is crucial for enhancing the understanding of the readers. To address this, we have made improvements to the readability of the paragraph that introduces copula functions. Additionally, we have incorporated the statistical concept of copulas within the BMA approach in Section 2.2, as suggested, and referenced the Cop-BMA methodological steps outlined by Madadgar and Moradkhani (2014) and Abbaszadeh et al. (2022a), which were employed in our study. The modified text and equations added reads as follows:

   *"Copulas are functions in the unit cube, which can link multi-dimensional distributions to their one-dimensional marginals (Sklar, 1959), they provide a flexible and powerful tool for modelling the dependency structure between variables, regardless of their individual marginal distributions and model dependency. This is particularly valuable in scenarios where the relationships between variables are complex and may not follow a simple linear pattern. Cop-BMA modifies the BMA predictive distribution through relaxing the assumption on parametric posterior distribution $g(y^t|M_i^t, \sigma_i^2)$ replaced with a group of multivariate copula functions. Multiple copula functions have been applied to postprocess hydrological forecasts (Abbaszadeh et al., 2022a; He et al., 2018; Madadgar et al., 2014; Madadgar and Moradkhani, 2014), and are used in this study for the estimation of water surface elevation posterior distribution. Equation 1 is modified to incorporate copula functions replacing the posterior distribution $p(y^t|M_i^t, Y)$ following the procedure from Abbaszadeh et al. (2022a). Supported*

*by Sklar theorem copulas can express the joint behaviour among correlated variables through their marginal CDFs in equation 3.*

$$P(x_1, \dots, x_n) = C[P(x_1, \dots, P(x_n)] = C(u_1, \dots u_n) \tag{3}$$

*where $C$ is the Cumulative Distribution Function (CDF) of the copula and $P(x_i)$ is the marginal distribution of $x_i$ denoted as $u_i$ for the interval [0, 1]. Using the PDF of copula, the joint probability density function of the variables involved can be defined as follows:*

$$p(x_1, \dots, x_n) = c(u_1, \dots, u_n) \prod_{i=i}^{n} p(x_i) \tag{4}$$

*The conditional probability distribution of $x_1$ given $x_2$ is defined in equation 5:*

$$p(x_1|x_2) = \frac{p(x_1, x_2)}{p(x_2)} \tag{5}$$

*Considering the copula joint probability from equation 4, the equation 5 can be expressed as:*

$$p(x_1|x_2) = \frac{p(x_1, x_2)}{p(x_2)} = \frac{c(u_1, u_2) \cdot p(x_1) \cdot p(x_2)}{p(x_2)} = c(u_1, u_2) \cdot p(x_1) \tag{6}$$

*Since $u_1$ and $u_2$ are the observations ($y^t$) and simulations ($M_k^t$) respectively, the posterior distribution in equation 1 is replaced with the conditional probability distribution from equation 6 as:*

$$p(y^t|M_1^t, M_2^t, \dots, M_k^t, Y) = \sum_{i=1}^{k} w_i \cdot p(y^t|M_i^k, Y) = \sum_{i=1}^{k} w_i \cdot c\left(u_{y^t}, u_{M_i^t}\right) \cdot p(y^t) \tag{7}$$

*Where $c(u_{y^t}, u_{M^t})$ represents the PDF of the copula function.* To estimate weight $w_i$, it is required to maximize the log likelihood function of the vector of parameter $\theta = \{w_i, i = 1, \dots, k\}$ as:

$$l(\theta) = \log\left(\sum_{i=1}^{k} w_i \sum_{t=1}^{T} c\left(u_{y^t}, u_{M_i^t}\right) \cdot p(y^t)\right) \tag{8}$$

Besides Section 2.2, the choice of copula in Section 4.1, line 329, is not convincing. It should be argued why only three distinct copula functions are evaluated and how that decision to restrict it to these three took place (one could imagine extreme value copulas or the Normal copula to be a candidate as well). For instance, whether there are certain characteristics that are sought in terms of attainable dependence, dimensionality, etc. Also, please detail the fitting process for evaluation on line 330 – it is unclear. The authors could use a copula cross-validation criterion, for instance, to make their choice tractable and not dependent on the rest of the model.

**Response:** We initially focused on the three most representative Archimedean copulas (Frank, Gumbel, Clayton) within our analysis, as they have been commonly used in previous hydrologic studies (Abbaszadeh et al., 2022; He et al., 2018). In response to the reviewer's comment, we expanded the scope of copulas considered to also include the Elliptic copulas (Gaussian and t-Student), as evaluated in the original COP-BMA paper (Madadgar & Moradkhani, 2014). To ensure robust copula selection, we employed the Akaike Information Criterion (AIC), which serves as a measure of the relative quality of statistical models. While this aspect was not previously mentioned in the article, we have incorporated it into the manuscript in Section 4.1 to provide details regarding our methodology. Furthermore, in accordance with the reviewer's suggestion, we conducted copula cross-validation using the criterion (xv-CIC) proposed by Grønneberg and Hjort (2014) employing a fold $k = 10$, yielding consistent conclusions with the AIC results for copula selection, which is also mentioned. Table S2 in supplementary data is added with results from criterion testing. We have updated the results section in the manuscript and included Table 4 to summarize the copulas selected per QPEs and cluster analyzed. The text reads as follows:

*"In this study, five distinct copula functions are evaluated, Gumbel, Clayton, and Frank from the class of Archimedean copulas, and Gaussian and t-Student from Elliptical group. Copulas are constructed and evaluated using the marginals distributions of the observed data and each of the precipitation products modelling results of water surface elevation per cluster as $c(u_y, u_{M_k})$. Fitting and selection process was conducted using Akaike Information Criterion (AIC) and copula cross-validation criterion (xv-CIC) (Grønneberg and Hjort, 2014) using copula package implemented in R (Hofert et al., 2023), where the copula fit with lowest value of AIC and higher xv-CIC was selected. Table 4 shows the selected copulas for the seven QPEs evaluated in HR2D simulations over the three clusters. Calculated values for AIC and xv-CIC are presented in Table S2 in the supplementary material."*

*Table 4. Summary of copula fitting results per cluster for each precipitation product used in the HEC-RAS 2D model simulations.*

| Precipitation Product | Transitional cluster | Upper cluster | Coastal cluster |
|---|---|---|---|
| CMORPH | Gumbel | Gumbel | t-Student |
| Daymet | Gumbel | Gumbel | t-Student |
| ERA5 | Gumbel | Gaussian | t-Student |
| IMERG | Gumbel | Gumbel | t-Student |
| NCEP Stage IV | Gumbel | Gumbel | t-Student |
| MRMS | Gumbel | Gumbel | t-Student |
| NLDAS | Gumbel | Gumbel | t-Student |

2. **Validation:** A substantial part of the manuscript aims to validate the approach by comparing the model prediction with the inundation level at certain stations and comparing these against single data-source predictions. As the COP-BMA nests all other single models, it is no surprise that it outperforms the others by design. A better comparison might be not against the other models but against a non-copula BMA. An alternative would be to omit one data source (likely the best performing rain gauge data) and to make the horserace of the COP-BMA against that single model. This comparison has additional implication as it would be a remote sensing data vs. rain gauge data comparison.

**Response:** We agree that analyzing QPEs against rain gauges and observations is a more reasonable approach than initially implemented. This approach provides the basis for comparing remote sensing QPE products, their COP-BMA integration and rain gauge data. In response, we have made modifications to the manuscript to reflect this adjustment:

- Rain gauge data has been excluded from the Global Cop-BMA analysis, and this exclusion has been reflected in the updated version of Table 3.
- To summarize the results more effectively and enhance clarity, we have generated a new Figure 6 to compare time-series of average error to observations per cluster. This figure compares errors generated by individual QPEs, Global Cop-BMA, and rain gauge simulations. Discussion of these results has been updated accordingly in the manuscript in Section 4.1. The new Figure 6 and modified text reads as follows:

[Figure]

***Figure 6. Averaged error time series of validation stations per cluster of simulated water surface elevation (WSE) by the HEC-RAS 2D model using QPE datasets and Global Cop-BMA approach results (black) against observed WSE during Hurricane Harvey.***

*"The averaged error of simulations using different QPEs against observations from validation stations are shown in Figure 6, featuring rain gauges simulation errors and estimations for Global Cop-BMA approach per cluster. Notably, this method exhibits better results in its responses to different precipitation products and clusters, leading to an enhanced accuracy in water level estimations, particularly during peak periods compared to the range of modelling water surface elevation outputs from the analyzed QPE such as Daymet or ERA5 which exhibit larger averaged errors. This demonstrates the Cop-BMA's capability to generate results that closely correspond to the observed values at the validation stations. It is important to highlight that if all models consistently overestimate or underestimate, Global Cop-BMA may not lead to significant improvement in the result (e.g., NOAA 8770613, USGS 08074710, USGS 08072050 in Figure 4; and Coastal cluster in Figure 6). Despite its advanced weighting mechanism, Global Cop-BMA's effectiveness relies on the diversity and accuracy of the model ensemble. Therefore, while it enhances the integration of diverse model outputs, its capability to improve results may be limited when all models exhibit similar differences compared to the observations at certain sections of the hydrograph".*

The modification implies changes to evaluation of new weights distribution per cluster. We updated the weights discussion in Section 4.1 as follows:

*"As depicted in Figure 7, the distribution of weights per cluster exhibits greater variability. In the transitional cluster, CMORPH, Daymet, ERA5 and NCEP Stage IV have weights below 0.1, as these four products generated underestimated responses in the hydrographs for most stations within this cluster. Weights center around the precipitation from MRMS, IMERG and NLDAS QPEs. Within the upper cluster, a different weights distribution among the QPEs is observable, with minimal influence from CMORPH, Daymet, MRMS and NLDAS QPEs. A higher difference is observable the three more dominant QPE, where NCEP Stage IV has a weight of 0.663 compared to the 0.144 of IMERG and 0.174 of ERA5 QPEs. For the Coastal cluster, precipitation from Stage IV QPE also holds the greatest weight (0.753) compared to the rest of QPEs which hold weight values below 0.1. Within this cluster, minimal discernible differences exist between QPEs water surface elevation results for the stations, as seen in Figure 4 (NOAA stations 8771013 and 8770613) and in Figure 6."*

- Figures 7, 8, and 9 have been updated to reflect the new weights calculation after excluding rain gauge data in the Cop-BMA analysis and incorporating Elliptical copula functions within the fitting process as discussed previously. These updates ensure that the figures accurately represent the latest calculations and findings.

- Section 4.1 was complemented and modified to be consistent with the change suggested.

3. **Implication:** As already indicated in the previous point raised, the manuscript lacks the precise quantification or discussion of the proposed methodology's advantages and applications. For instance, staying with the case of Hurricane Harvey: what are the computational costs and runtime of the model? Could it be employed in risk assessment with rain-on-grid data forecasts? How much better is it compared to a non-COP BMA? While these questions are for illustration only, the discussion in the manuscript could aim to assess the methodology's benefit in a broader context.

**Response:** Thank you for your comment. In response, we have made the following revisions to strengthen the implications and broader impacts of our study:

- Section 5 has been edited to discuss the potential use of alternative precipitation forecast datasets and the application of our framework for operational forecasting purposes. This discussion expands the scope of our study and highlights its relevance for real-world applications and future areas of improvements. We have edited the text in the Discussions and Conclusions section as follows:

*"One advantage of our proposed framework is its flexibility allowing for the use of alternative precipitation products to enhance model simulations. For instance, this framework can be implemented for operational forecasting purposes where the Quantitative Precipitation Estimations (QPEs) utilized in this study can be replaced with Quantitative Precipitation Forecasts (QPF) from numerical weather prediction models such as High-Resolution Rapid Refresh (HRRR), North American Mesoscale Forecast System (NAM), Global Forecast System (GFS), European Centre for Medium-Range Weather Forecasts (ECMWF) among others".*

- Computational resources and computational time required for the hydrodynamic modeling task have been addressed in Section 3.1, providing readers with insight into the practical aspects of implementing the proposed methodology. We consider the computational times (~7 hours) to be adequate for the application of the proposed methodology, and future research efforts will focus on utilizing the Linux version of HEC-RAS 2D on High-Performance Computing (HPC) systems. Added text reads as follows:

  *"It is worth mentioning that our simulations were performed on a desktop computer with an Intel Core i7-7700 CPU @ 3.60GHz and 32GB RAM memory, averaging about seven hours per simulation for the time window."*

- The improvements achieved using the Cop-BMA approach in terms of overall performance have been integrated into Section 4.1. This section provides a detailed analysis of the enhanced performance resulting from the application of the Cop-BMA methodology. The modified text in the manuscript describing the Cop-BMA results reads as follows:

  *"Figure 8 provides a comprehensive overview of collective performance metrics of the HR2D model across the seven QPE simulations, rain gauges simulation, and the Global Cop-BMA multi-modelling for the seven QPEs evaluated at 30 validation stations over the 11-day simulation period. In general, the inundation modelling driven by different products consistently exhibits NSE performance with mean values ranging between 0.695 and 0.846. In terms of KGE performance, the interquartile ranges for QPEs display broader ranges, and the medians for Daymet and ERA5 products fall below 0.8 in contrast to other simulations.*
  *Notably, the Cop-BMA approach exhibits slightly higher performance metrics compared to the QPE products, NSE has an average of 0.858 and its total variability is lower compared to single precipitation products. KGE metric has a similar result with an average value of 0.852. The Averaged RMSE for Cop-BMA is 0.561m which is smaller than all the single QPE except for the rain gauges simulation which is only 3 centimetres lower. The averaged MBE for single QPEs ranged between -0.018 and 0.23m, while the Global Cop-BMA method results in an averaged value of 0.049m. Among individual products, the rain gauge outperforms all spatially distributed precipitation datasets and comes closest to matching the performance of Cop-BMA method. This highlights that reanalysis gridded precipitation products may have higher errors when compared to in-situ rain observations and allows Global Cop-BMA to generate QPE post-processed results that are closer to modelling results with observed precipitation from rain gauges. This methodology could be replicated in areas where measured precipitation is not available and obtain better performance metrics accounting for the uncertainties from this input".*

Minor remarks:

- Line 52: please introduce the HEC-RAS 2D when first mentioning it, briefly.
  **Response:** Description of HEC-RAS 2D model was added in Section 1. Text reads as follows:

*"Among various hydrodynamic models, the Hydrologic Engineering Center's River Analysis System (HEC-RAS) developed by the United States Army Corps of Engineers (USACE, 2022). It has the capability to simulate flooding conditions allows in both 1D and 2D."*

- Line 83: please discuss the advantages and disadvantages of deterministic vs. probabilistic approach in this setting.
  **Response:** Description and discussion of deterministic and probabilistic approaches for flood mapping was added to the manuscript along with references from Di Baldassarre et al. (2010); Bates et al. (2004) and Merwade et al. (2008) to the readers interested in the topic. We have included your suggestion in the revised manuscript as follows:

  *"On the deterministic front, the numerical results of the HEC-RAS 2D 6.3.1 hydrodynamic model, incorporating RoG, are evaluated to best describe the hydrodynamic behaviour of rivers, coastal and floodplain processes with a computationally affordable model. In parallel, a probabilistic approach is employed to use eight distinct precipitation products as forcing data to the hydrodynamic model to estimate an ensemble of flood extent and water depth in response to this hurricane-induced flood event. The deterministic approach provides a single representation of flood extents and depths based on predefined inputs and parameters, offering a clear understanding of potential inundation scenario evaluated. However, it fails to adequately capture the uncertainty associated with flood modelling, potentially leading to underestimation or overestimation of flood extents in other scenarios considering highly sensitive input parameters, which can impact the accuracy of results (Di Baldassarre et al., 2010; Bates et al., 2004).*
  *Probabilistic flood inundation mapping incorporates probabilistic techniques to assess and quantify uncertainty, providing a more comprehensive understanding of the range of potential flood outcomes and associated risks. It allows the integration of different datasets and input values, accommodating a wider range of initial and boundary conditions, and improving the robustness of flood predictions (Merwade et al., 2008; Di Baldassarre et al., 2010). Often this approach requires conducting numerous simulations to assess parameter uncertainty, leading to a substantial consumption of computational resources. Consequently, there is a preference for utilizing models that make substantial flow assumptions to conduct these simulations more efficiently and reduce computational costs."*

- Line 123: please precisely state which variables are all subject of BMA.
  **Response**: The water surface elevation was the variable of interest. This information was added.
- Line 133: "In other worlds..."
  **Response:** Edited to "In summary".
- Line 148: Is the copula function bivariate? Please introduce the concept of a copula function here.
  **Response:** Copulas evaluated in this study are bivariate. Description of copula functions and formulation within BMA approach were added to the manuscript in Section 2.2 as mentioned in Comment 1.

Additional references included to manuscript:

**Bates, P. D**., Horritt, M. S., Aronica, G., and Beven, K.: Bayesian updating of flood inundation likelihoods conditioned on flood extent data, Hydrological Processes, 18, 3347–3370, https://doi.org/10.1002/hyp.1499, 2004.

**Di Baldassarre**, G., Schumann, G., Bates, P. D., Freer, J. E., and Beven, K. J.: Flood-plain mapping: a critical discussion of deterministic and probabilistic approaches, Hydrological Sciences Journal, 55, 364–376, https://doi.org/10.1080/02626661003683389, 2010.

**Grønneberg, S**. and Hjort, N. L.: The Copula Information Criteria, Scandinavian J Statistics, 41, 436–459, https://doi.org/10.1111/sjos.12042, 2014.

**Merwade, V**., Olivera, F., Arabi, M., and Edleman, S.: Uncertainty in Flood Inundation Mapping: Current Issues and Future Directions, J. Hydrol. Eng., 13, 608–620, https://doi.org/10.1061/(ASCE)1084-0699(2008)13:7(608), 2008

**Sklar, M.:** Fonctions de répartition à N dimensions et leurs marges, Annales de l'ISUP, 229–231, 1959.

Thank you again for your constructive comments.

Sincerely,

Francisco J. Gomez, Corresponding Author
Center for Complex Hydrosystems Research (CCHR)
Department of Civil, Construction and Environmental Engineering
The University of Alabama

---

## Author Comment (AC2)

**Response to Referee 2 Comments on:**

**Probabilistic Flood Inundation Mapping through Copula Bayesian Multi-Modelling of Precipitation Products**

Francisco Javier Gomez, Keighobad Jafarzadegan, Hamed Moftakhari, and Hamid Moradkhani

MS. Ref. No.: NHESS-2024-26
Natural Hazards and Earth System Sciences

We appreciate the time spent in the review process of the article manuscript. The feedback provided for the submission is of high relevance in order to improve the quality of our manuscript. Enclosed, please find the revised manuscript with all the necessary corrections implemented in accordance with the comments and suggestions in tracked and cleaned versions. The following are point-by-point answers, in blue color, to the comments made by the referee.

**Answers to referee #2:**

1. Although the technical details of the copula functions have been presented in the literature, it will be helpful for readers to catch the meanings of new variables presented in this article if these variables are illustrated clearly. For example, what are the terms in the copula function on the right-hand side of Equation (3)?

   **Response:** To address this comment, we have further extended the description of copulas for better understanding of the readers. Specifically, in response to the suggestions provided, in Section 2.2 we have expanded the description of key terms in equations, the formulation of copulas and implementation as Cop-BMA. We added references of original formulation from Sklar (1959) and previous Cop-BMA methodology from Madadgar et al., (2014) and Abbaszadeh et al. (2022) to provide readers with additional context and theoretical explanations. These additions aim to facilitate a deeper understanding and address questions that may arise. The modified text and equations added reads as follows:

   *"Copulas are functions in the unit cube, which can link multi-dimensional distributions to their one-dimensional marginals (Sklar, 1959), they provide a flexible and powerful tool for modelling the dependency structure between variables, regardless of their individual marginal distributions and model dependency. This is particularly valuable in scenarios where the relationships between variables are complex and may not follow a simple linear pattern. Cop-BMA modifies the BMA predictive distribution through relaxing the assumption on parametric posterior distribution $g(y^t|M_i^t, \sigma_i^2)$ replaced with a group of multivariate copula functions. Multiple copula functions have been applied to postprocess hydrological forecasts (Abbaszadeh et al., 2022a; He et al., 2018; Madadgar et al., 2014; Madadgar and Moradkhani, 2014), and are used in this study for the estimation of water surface elevation posterior distribution. Equation 1 is modified to incorporate copula functions replacing the posterior distribution $p(y^t|M_i^t, Y)$ following the procedure from Abbaszadeh et al. (2022a). Supported*

*by Sklar theorem copulas can express the joint behaviour among correlated variables through their marginal CDFs in equation 3.*

$$P(x_1, \dots, x_n) = C[P(x_1, \dots, P(x_n)] = C(u_1, \dots u_n) \tag{3}$$

*where $C$ is the Cumulative Distribution Function (CDF) of the copula and $P(x_i)$ is the marginal distribution of $x_i$ denoted as $u_i$ for the interval [0, 1]. Using the PDF of copula, the joint probability density function of the variables involved can be defined as follows:*

$$p(x_1, \dots, x_n) = c(u_1, \dots, u_n) \prod_{i=i}^{n} p(x_i) \tag{4}$$

*The conditional probability distribution of $x_1$ given $x_2$ is defined in equation 5:*

$$p(x_1|x_2) = \frac{p(x_1, x_2)}{p(x_2)} \tag{5}$$

*Considering the copula joint probability from equation 4, the equation 5 can be expressed as:*

$$p(x_1|x_2) = \frac{p(x_1, x_2)}{p(x_2)} = \frac{c(u_1, u_2) \cdot p(x_1) \cdot p(x_2)}{p(x_2)} = c(u_1, u_2) \cdot p(x_1) \tag{6}$$

*Since $u_1$ and $u_2$ are the observations $(y^t)$ and simulations $(M_k^t)$ respectively, the posterior distribution in equation 1 is replaced with the conditional probability distribution from equation 6 as:*

$$p(y^t|M_1^t, M_2^t, \dots, M_k^t, Y) = \sum_{i=1}^{k} w_i \cdot p(y^t|M_i^k, Y) = \sum_{i=1}^{k} w_i \cdot c\left(u_{y^t}, u_{M_i^t}\right) \cdot p(y^t) \tag{7}$$

*Where $c(u_{y^t}, u_{M^t})$ represents the PDF of the copula function. To estimate weight $w_i$, it is required to maximize the log likelihood function of the vector of parameter $\theta = \{w_i, i = 1, \dots, k\}$ as:*

$$l(\theta) = \log\left(\sum_{i=1}^{k} w_i \sum_{t=1}^{T} c\left(u_{y^t}, u_{M_i^t}\right) \cdot p(y^t)\right) \tag{8}$$

2. As shown in Equation (4), the form of the likelihood function (a product of the probability density at different time steps) is valid when the temporal predictions are independent. But in this study, we may not assume the "hourly" output to be independent of each other. So what are the potential impacts of autocorrelation of the target variables, y, on the Cop-BMA results?

   **Response:** We consider that hydrodynamic simulations resulting from the different QPEs are independent of each other as these products come from different sources and methodologies, with both different spatial and temporal resolutions. For readers interested in a more detailed analysis, we refer to the original BMA and Cop-BMA research article by Madadgar and Moradkhani (2014) in the manuscript. Additionally, equation 4 (equation 8 in recent edited version) was edited as the summation over time term was typed in an incorrect position as shown in the comment 1.

3. It is not clear how to transform the rainfall into runoff in the HEC-RAS 2D model. Also, it seems to be unfair to compare the performance of different precipitation products since the infiltration process was not considered in the hydrodynamic modeling process. In addition, in Line 453, why do the infiltration processes mainly impact the "initial" water surface elevation results? How about the "continuous" loss during the flood event?

   **Response:** We appreciate the reviewer's feedback regarding infiltration processes, and we acknowledge the importance of considering these hydrological processes in hydrodynamic models. It is commonly assumed, especially for events like Hurricane Harvey in highly urbanized areas such as Houston, that the soil was completely saturated due to previous rainfall and the percentage of impervious cover is significant. While this assumption simplifies the analysis by neglecting infiltration processes, it may not fully capture the dynamic nature of soil properties and runoff transformation in hydrodynamic simulations as mentioned in the manuscript. In light of this, we recognize the need to explore infiltration processes more comprehensively in future research. We have outlined this aspect in Section 5 for the benefit of the readers as future research topics. This includes testing different infiltration methods

directly within the HR2D model, such as Deficit and Constant, SCS Curve Number, and Green-Ampt, across various storm events in rural areas with more diverse land cover. The additional text reads as follows:

*"The HEC-RAS model can also incorporate the impact of infiltration during flood events. This involves testing various infiltration methods, such as Deficit and Constant, SCS Curve Number, and Grenn-Ampt, across different storm events in rural areas with diver land cover."*

4. Manning values are important parameters in flood modeling and the values will change with the water depth. As presented in Line 192, the Manning values "are further adjusted during the calibration period, 7 days before the occurrence of Hurricane Harvey", would it be better or necessary to calibrate the roughness parameters based on a similar flood event?
   **Response:** Previous research conducted by our research group, as outlined in Muñoz et al. (2022) focused on the estimation of Manning roughness coefficients using Latin Hypercube Sampling (LHS) within the study area considering the extreme water levels generated at the peak of Hurricane Harvey for calibration procedure. The values obtained from this research were thoroughly tested and, where necessary, slightly modified to suit the specific requirements of the current study using HR2D. We cite this previous work as a guiding reference to the readers for the methodology employed and the selection of Manning coefficients in our study. Modified text in Section 3.1 reads as follows:

   *"In a previous research conducted by Muñoz et al. (2022), they used Latin Hypercube Sampling and tested various Manning roughness values for different land cover categories during Hurricane Harvey event. We use their calibrated parameters as a reference for HR2D model setup. These values are slightly adjusted during the calibration period, 7 days before the occurrence of Hurricane Harvey".*

5. For the application of Cop-BMA, it is like a trial-and-error procedure to select an appropriate marginal distribution and a copula function for the target variable. Is there any general guidance or suggestion for interested readers if they want to apply the framework to the other areas or variables?
   **Response:** We appreciate the reviewer's constructive suggestion. To provide clarity and guidance to interested readers, we have incorporated a description of the criteria used in the selection process in Section 4.1 of the manuscript. The selection of marginal distribution is based on minimizing the sum of squared error (SSE) using the Maximum Likelihood Estimation (MLE). Modified section reads as follows:

   *"Parameter estimation for each distribution is performed using the Maximum Likelihood Estimation (MLE) technique. To identify the most suitable marginal distribution, the sum of squared errors (SSE) is employed to facilitate the selection process, choosing the distribution that provide the lower SSE value."*

   To select the appropriate copula, we calculated the Akaike information criterion (AIC) and cross-validation criterion (xv-CIC) for each copula and then chose the one with the lowest AIC and highest xv-CIC. Details of this analysis have been included in Table S2 and can be found in the supplementary materials. We recommend interested readers follow this procedure when applying the framework to other studies. Added text to Section 4.1 reads as follows:

*"Fitting and selection process was conducted using Akaike Information Criterion (AIC) and copula cross-validation criterion (xv-CIC) (Grønneberg and Hjort, 2014) using copula package implemented in R (Hofert et al., 2023), where the copula fit with lowest value of AIC and higher xv-CIC was selected. Table 4 shows the selected copulas for the seven QPEs evaluated in HR2D simulations over the three clusters. Calculated values for AIC and xv-CIC are presented in Table S2 in the supplementary material."*

6. Some cases in Figures 4 and 6 show that if all the members in the precipitation ensembles consistently overestimated (e.g., NOAA 8770613 and USGS 08074710) or underestimated (e.g., USGS 08072050) the peak WSE, Global Cop-BMA did not help at all. Any comments on that?

    **Response:** These discrepancies primarily arise from errors in the model structure and the parameterization of the hydrodynamic model. Factors such as assumptions within the governing equations of the HR2D model, infiltration methods, and the absence of full bathymetry datasets are identified as significant contributors to these differences in WSE values. In Section 4 of the manuscript, we discussed this issue. In our ongoing research, we aim to address these discrepancies by exploring alternative numerical models for evaluation and testing purposes. By incorporating additional modeling frameworks, we seek to refine and improve the accuracy of our simulation results.

    Additionally, when we use the Cop-BMA methodology, the highest weight is assigned to the product that provides the closest results to the observations. However, in cases where all model members generate overestimated or underestimated results, there may not be a meaningful improvement. We incorporate this important remark of Global Cop-BMA methodology in Section 4.1, the added text reads as follows:

    *"It is important to highlight that if all models consistently overestimate or underestimate, Global Cop-BMA may not lead to significant improvement in the result (e.g., NOAA 8770613, USGS 08074710, USGS 08072050 in Figure 4; and Coastal cluster in Figure 6). Despite its advanced weighting mechanism, Global Cop-BMA's effectiveness relies on the diversity and accuracy of the model ensemble. Therefore, while it enhances the integration of diverse model outputs, its capability to improve results may be limited when all models exhibit similar differences compared to the observations at certain sections of the hydrograph."*

7. Line 20 and Line 382, could you provide quantitative results to measure the degree of improvement due to the application of the Cop-BMA approach?

    **Response:** We appreciate this comment which shows general improvements in the proposed methodology. We have added this information to the abstract, providing an overview of the performance achieved through the application of the Cop-BMA approach specially with NSE metric results. Additionally, in Section 4.1 of the manuscript, detailed results and NSE, KGE, RMSE and MBE performance metrics from the utilization of the Cop-BMA approach have been elaborated as discussion of the results obtained in the boxplots plotted in Figure 8. This analysis offers a comprehensive examination of the quantitative outcomes derived from our study, thereby facilitating a more thorough evaluation of the effectiveness and impact of the Cop-BMA methodology. The modified text in the manuscript reads as follows:

*"Figure 8 provides a comprehensive overview of collective performance metrics of the HR2D model across the seven QPE simulations, rain gauges simulation, and the Global Cop-BMA multi-modelling for the seven QPEs evaluated at 30 validation stations over the 11-day simulation period. In general, the inundation modelling driven by different products consistently exhibits NSE performance with mean values ranging between 0.695 and 0.846 In terms of KGE performance, the interquartile ranges for QPEs display broader ranges, and the medians for Daymet and ERA5 products fall below 0.8 in contrast to other simulations. Notably, the Cop-BMA approach exhibits slightly higher performance metrics compared to the QPE products, NSE has an average of 0.858 and its total variability is lower compared to single precipitation products. KGE metric has a similar result with an average value of 0.852. The Averaged RMSE for Cop-BMA is 0.561m which is smaller than all the single QPE except for the rain gauges simulation which is only 3 centimetres lower. The averaged MBE for single QPEs ranged between -0.018 and 0.23m, while the Global Cop-BMA method results in an averaged value of 0.049m."*

8. As discussed by the authors, the final flood inundation maps could not be validated effectively because of the scarcity of spatial observed data. Is it possible that the performance of a model member in the ensemble would be better than that of Cop-BMA in terms of the inundation extents, even though its performance in the WSE comparison at one gauge location is not the best?

   **Response**: It is worth noting that while a flood inundation map provided by a single QPE may potentially exhibit greater accuracy compared to one generated by Cop-BMA, the primary advantage of using Cop-BMA lies in its ability to generate probabilistic flood inundation maps while considering uncertainties associated with various QPE sources. Additionally, the QPE offering the highest accuracy is not consistently a single product; it may vary across different case studies and flood events. Therefore, employing a BMA-based approach could be a viable strategy to achieve high accuracy while addressing sources of uncertainty. Additional text in Section 4.1 reads as follows:

   *"The validation tasks were primarily focused on assessing the performance of model outputs at validation stations, as depicted in Figure 5. This approach enabled us to calculate the performance metrics of WSE over a well-distributed network of stations with remarkable temporal resolution. Data collected from these validation stations sufficiently capture the hydrograph behaviour within the study domain and enables us to quantify flood extents in a probabilistic manner using the HR2D model incorporated with the Cop-BMA method. It is worth noting that while a flood inundation map provided by a single QPE may potentially exhibit greater accuracy compared to one generated by Cop-BMA, the primary advantage of using Cop-BMA lies in its ability to generate probabilistic flood inundation maps while considering uncertainties associated with various QPE sources. Additionally, the QPE offering the highest accuracy is not consistently a single product; it may vary across different study cases and flood event characteristics. Therefore, employing a BMA-based approach could be a viable strategy to achieve high accuracy while accounting for uncertainties."*

9. Minor Issues

- Line 42-43, in BMA applications, I think it is the conditional PDF (the second term on the right-hand side of Equation (1)) rather than the "data" that is assumed to follow a Gaussian distribution. In other words, the pattern of model residuals follows a Gaussian distribution in BMA.
  **Response:** Usually, the conditional PDF of the data is assumed to follow a Gaussian distribution in hydrologic-hydrodynamic applications. This phrase was adjusted to be more concise.
- It would be better if more information can be added to Table 1 or Figure 2. For example, the temporal resolution of discharge and WSE data, the start and end date of the simulation period, indicating which stations are used as boundary conditions and validation, etc. Also, the information of USGS 08074710 was not included in Table 1.
  **Response:**
  - Section 3.1 was complemented with a brief description of simulation time window in HR2D model and mention the hourly outputs. Information is incorporated to the manuscript as follows:

    *"For unsteady flow analysis in HR2D setup, an hourly simulation time window is defined between August 16/2017 to September 3/2017."*

  - Within the description of BCs in Section 3.1 the hourly temporal resolution of these inputs was incorporated, except for QPEs which are explained in Table 2.
  - We include in the Supplementary data document Table S1 with the stations used for validation with details and referred in the manuscript.
  - Figure 5 in Section 4.1 shows stations used for validation during the hurricane event, station USGS 08074710 is part of this validation set, not as BC.
- Line 73: two brackets were used for the reference. **Response:** Two brackets were deleted.
- Please add the units of SSE in Table 3. **Response:** Units added to table.

Additional references included to manuscript:

**Grønneberg, S**. and Hjort, N. L.: The Copula Information Criteria, Scandinavian J Statistics, 41, 436–459, https://doi.org/10.1111/sjos.12042, 2014.
**Sklar, M.:** Fonctions de répartition à N dimensions et leurs marges, Annales de l'ISUP, 229–231, 1959.

Thank you again for your constructive comments.

Sincerely,

Francisco J. Gomez, Corresponding Author
Center for Complex Hydrosystems Research (CCHR)
Department of Civil, Construction and Environmental Engineering
The University of Alabama

---

## Author Response (AR2)

MS. Ref. No.: NHESS-2024-26
Natural Hazards and Earth System Sciences

We appreciate the comments provided by the reviewer and the editor for improving the quality of the last version of the manuscript. The following are point-by-point answers to the comments in blue color:

**Comments from Referee #2 (RC2):**

1.The corrected log-likelihood function, Eqn. (8), in your revised manuscript is slightly different from the original version in the BMA literature (Raftery et al., 2005), in which the prediction errors are assumed to be independent over time. Even if the final BMA estimates may be not significantly different due to the different forms of likelihood functions, it would be better to justify the assumption you have made and check the results again.

Response: Thank you for drawing attention to this distinction. We acknowledge that both versions of likelihood functions have been extensively employed in the literature. As correctly pointed out by the reviewer, the original version, proposed by Raftery et al. (Dong et al., 2013; Liu & Merwade, 2018; Parrish et al., 2012), assumes independent probability functions over time, resulting in different weights estimated at each time step. However, the version implemented by us assigns a single weight to the entire simulation period (Abbaszadeh et al., 2022; Duan et al., 2007; He et al., 2018; Madadgar & Moradkhani, 2014).

In response to the reviewer's suggestion, we investigated the sensitivity of our results to both likelihood functions. The table below illustrates the performance metrics of Cop-BMA results using both versions of the likelihood function. As evident, the performance does not significantly vary between these two versions. Consequently, we have opted to retain this implementation, as it yields results comparable to the original version and aligns with the broader usage in the literature.

| Metric | Averaged value | |
|---|---|---|
| | Eq. in article | Raftery et al. |
| NSE | 0.8585 | 0.8581 |
| KGE | 0.8527 | 0.8520 |
| RMSE | 0.5617m | 0.5696m |
| MBE | 0.049m | 0.0535m |

2. There might be a typo in Eqn. (3) in your revised manuscript. The right parentheses of "P(x_1" in the middle term is missing.
**Response:** The equation was corrected.

**Comments from Editor:**

I kindly ask you to increase the font size of the labels in Figures 3, 4, 6 and 8. Figure 9 is missing a scale bar and a north arrow or grid.

**Response:** Figures 3, 4, 6 and 8 were modified by increasing the font size of labels. Figure 9 was modified by adding a north arrow and a scale bar to be consistent with other maps in the manuscript.

References

Abbaszadeh, P., Gavahi, K., Alipour, A., Deb, P., & Moradkhani, H. (2022). Bayesian Multi-modeling of Deep Neural Nets for Probabilistic Crop Yield Prediction. *Agricultural and Forest Meteorology*, *314*, 108773. https://doi.org/10.1016/j.agrformet.2021.108773

Dong, L., Xiong, L., & Yu, K. (2013). Uncertainty Analysis of Multiple Hydrologic Models Using the Bayesian Model Averaging Method. *Journal of Applied Mathematics*, *2013*, 1–11. https://doi.org/10.1155/2013/346045

Duan, Q., Ajami, N. K., Gao, X., & Sorooshian, S. (2007). Multi-model ensemble hydrologic prediction using Bayesian model averaging. *Advances in Water Resources*, *30*(5), 1371–1386. https://doi.org/10.1016/j.advwatres.2006.11.014

He, S., Guo, S., Liu, Z., Yin, J., Chen, K., & Wu, X. (2018). Uncertainty analysis of hydrological multi-model ensembles based on CBP-BMA method. *Hydrology Research*, *49*(5), 1636–1651. https://doi.org/10.2166/nh.2018.160

Liu, Z., & Merwade, V. (2018). Accounting for model structure, parameter and input forcing uncertainty in flood inundation modeling using Bayesian model averaging. *Journal of Hydrology*, *565*, 138–149. https://doi.org/10.1016/j.jhydrol.2018.08.009

Madadgar, S., & Moradkhani, H. (2014). Improved Bayesian multimodeling: Integration of copulas and Bayesian model averaging. *Water Resources Research*, *50*(12), 9586–9603. https://doi.org/10.1002/2014WR015965

Parrish, M. A., Moradkhani, H., & DeChant, C. M. (2012). Toward reduction of model uncertainty: Integration of Bayesian model averaging and data assimilation. *Water Resources Research*, *48*(3). https://doi.org/10.1029/2011WR011116

Sincerely,

Francisco J. Gomez, Corresponding Author
Center for Complex Hydrosystems Research (CCHR)
Department of Civil, Construction and Environmental Engineering
The University of Alabama

---

## Author Response (AR3)

MS. Ref. No.: NHESS-2024-26
Natural Hazards and Earth System Sciences

We appreciate the comments provided by the editor for improving the quality of the last version of the manuscript. The following are point-by-point answers to the comments in blue color:

**Comments from review file validation**

Please ensure that the colour schemes used in your maps and charts allow readers with colour vision deficiencies to correctly interpret your findings. Please check your figures using the Coblis – Color Blindness Simulator (https://www.color-blindness.com/coblis-color-blindness-simulator/) and revise the colour schemes accordingly.

Response: We checked the color schemes for the figures as suggested. For Figure 9 we decided to remove the black grey basemap as it interferes with the data in the monochromatic view filter.

Sincerely,

Francisco J. Gomez, Corresponding Author
Center for Complex Hydrosystems Research (CCHR)
Department of Civil, Construction and Environmental Engineering
The University of Alabama